# Plugin Estimators for Selective Classification with Out-of-distribution Detection

**Harikrishna Narasimhan,  Aditya Krishna Menon,  Wittawat Jitkrittum,  Sanjiv Kumar**
Google Research
{hnarasimhan, adityakmenon, wittawat, sanjivk}@google.com

## Abstract

Real-world classifiers can benefit from optionally *abstaining* from predicting on samples where they have low confidence. Such abstention is particularly useful on samples which are close to the learned decision boundary, or which are outliers with respect to the training set. These settings have been the subject of extensive but disjoint study in the *selective classification* (*SC*) and *out-of-distribution* (*OOD*) detection literature. Recent work on *selective classification with OOD detection* (*SCOD*) has argued for the unified study of these problems; however, the formal underpinnings of this problem are still nascent, and existing techniques are heuristic in nature. In this paper, we propose new plugin estimators for SCOD that are theoretically grounded, effective, and generalise existing approaches from the SC and OOD detection literature. In the course of our analysis, we formally explicate how naïve use of existing SC and OOD detection baselines may be inadequate for SCOD. We empirically demonstrate that our approaches yields competitive SC and OOD detection trade-offs compared to common baselines.

## 1 Introduction

Given a training sample drawn i.i.d. from a distribution $\mathbb{P}_{in}$ (e.g., images of cats and dogs), the standard classification paradigm concerns learning a classifier that accurately predicts the label for test samples drawn from $\mathbb{P}_{in}$. However, in real-world deployment, one may encounter *out-of-distribution* (*OOD*) test samples, i.e., samples drawn from some $\mathbb{P}_{out} \neq \mathbb{P}_{in}$ (e.g., images of aeroplanes). *Out-of-distribution detection* is the problem of accurately identifying such OOD samples, and has received considerable recent study (Hendrycks and Gimpel, 2017; Lee et al., 2018; Hendrycks et al., 2019; Ren et al., 2019; Huang et al., 2021; Huang and Li, 2021; Thulasidasan et al., 2021; Wang et al., 2022a; Bitterwolf et al., 2022; Katz-Samuels et al., 2022; Wei et al., 2022; Sun et al., 2022; Hendrycks et al., 2022). An accurate OOD detector allows one to *abstain* from making a prediction on OOD samples, rather than making an egregiously incorrect prediction; this enhances reliability and trust-worthiness.

The quality of an OOD detector is typically assessed by its ability to distinguish in-distribution (ID) versus OOD samples. However, some recent works (Kim et al., 2021; Xia and Bouganis, 2022; Cen et al., 2023; Humblot-Renaux et al., 2024) argued that to accurately capture the real-world deployment of OOD detectors, it is important to consider distinguishing *correctly-classified ID* versus *OOD and misclassified ID* samples. Indeed, it is intuitive that a classifier not only abstain on OOD samples, but also abstains from predicting on "hard" (e.g., ambiguously labelled) ID samples which are likely to be misclassified. This problem is termed *unknown detection* (*UD*) in Kim et al. (2021), and *selective classification with OOD detection* (*SCOD*) in Xia and Bouganis (2022); we adopt the latter in the sequel. One may view SCOD as a unification of OOD detection and the *selective classification* (*SC*) paradigm (Chow, 1970; Bartlett and Wegkamp, 2008; El-Yaniv and Wiener, 2010; Cortes et al., 2016b; Ramaswamy et al., 2018; Ni et al., 2019; Cortes et al., 2023; Mao et al., 2024).

Both OOD detection and SC have well-established formal underpinnings, with accompanying principled techniques (Bitterwolf et al., 2022; Cortes et al., 2016b; Ramaswamy et al., 2018); however, by contrast, the theoretical understanding of SCOD is relatively nascent. Prior work on SCOD employs a heuristic design to construct the rejection rule (Xia and Bouganis, 2022). Specifically, given confidence scores for correct classification, and scores for OOD detection, the mechanism heuristically combines the two scores to decide whether to abstain on a sample. It remains

Table 1: Summary of two different settings for SCOD. Our goal is to learn a classifier capable of rejecting *both* out-of-distribution (OOD) and "hard" in-distribution (ID) samples. We present a plug-in estimator for SCOD, and apply it two settings: one with access to only ID data, the other with additional access to a unlabeled mixture of ID and OOD data (Katz-Samuels et al., 2022). In both cases, we reject samples by suitably combining scores that order samples based on selective classification (SC) or OOD detection criteria. The former setting leverages *any* off-the-shelf scores for these tasks, while the latter minimises a joint loss functions to estimate these scores.

| | **Black-box SCOD** | **Loss-based SCOD** |
|---|---|---|
| **Training data** | ID data only | ID + OOD data |
| **SC score** $s_{\mathrm{sc}}$ | Any off-the-shelf technique, e.g., maximum softmax probability (Chow, 1970) | Minimise (10), obtain $\max_{y \in [L]} f_y(x)$ |
| **OOD score** $s_{\mathrm{ood}}$ | Any off-the-shelf technique, e.g., gradient norm (Huang et al., 2021) | Minimise (10), obtain $s(x)$ |
| **Rejection rule** | Combine $s_{\mathrm{sc}}, s_{\mathrm{ood}}$ via (8) | |

unclear if there are settings where this approach may fail, and what the optimal combination strategy would look like. We aim to address these questions in this paper.

More concretely, we provide a statistical formulation for the SCOD problem, and derive the Bayes-optimal solution. Based on this solution, we propose a *plug-in* approach for SCOD: this takes confidence estimates for SC and density estimates for OOD detection, and optimally combines them to output a rejection decision. We then show case how our plug-in approach can be applied under two different assumptions on available data during training (Table 1). The first is the challenging setting where one has access to *only* ID data, and we leverage existing techniques for SC and OOD detection in a *black-box* manner. The second is the setting of Katz-Samuels et al. (2022), where one additionally has access to an unlabeled "wild" sample comprising a mixture of both ID and OOD data, and one can use techniques such as Thulasidasan et al. (2021); Bitterwolf et al. (2022) to *jointly* estimate scores for SC and OOD detection. In summary, our contributions are:

(i) We provide a statistical formulation for SCOD that unifies both the SC and OOD detection problems (§3), and derive the corresponding Bayes-optimal solution (Lemma 3.1), which combines scores for SC and OOD detection. Intriguingly this solution is a variant of the popular maximum softmax probability baseline for SC and OOD detection (Chow, 1970; Hendrycks and Gimpel, 2017), using a *sample-dependent* rather than constant threshold.

(ii) Based on the form of the Bayes-optimal solution, we propose a plug-in approach for SCOD (§4), and showcase how it can be applied to a setting with access to only ID data (§4.1), and the setting of Katz-Samuels et al. (2022) with access to a mixture of ID and OOD data (§4.2).

(iii) Experiments on benchmark image classification datasets (§5) show that our plug-in approach yields competitive classification and OOD detection performance at any desired abstention rate, compared to the heuristic approach of Xia and Bouganis (2022), and other common baselines.

## 2   BACKGROUND AND NOTATION

We focus on multi-class classification problems: given instances $\mathcal{X}$, labels $\mathcal{Y} \doteq [L]$, and a training sample $S = \{(x_n, y_n)\}_{n \in [N]} \in (\mathcal{X} \times \mathcal{Y})^N$ comprising $N$ i.i.d. draws from a *training* (or *inlier*) *distribution* $\mathbb{P}_{\mathrm{in}}$, the goal is to learn a classifier $h \colon \mathcal{X} \to \mathcal{Y}$ with minimal misclassification error $\mathbb{P}_{\mathrm{te}}(y \neq h(x))$ for a *test distribution* $\mathbb{P}_{\mathrm{te}}$. By default, it is assumed that the training and test distribution coincide, i.e., $\mathbb{P}_{\mathrm{te}} = \mathbb{P}_{\mathrm{in}}$. Typically, $h(x) = \operatorname{argmax}_{y \in [L]} f_y(x)$, where $f \colon \mathcal{X} \to \mathbb{R}^L$ scores the affinity of each label to a given instance. One may learn $f$ via minimisation of the *empirical surrogate risk* $\hat{R}(f; S, \ell) \doteq \frac{1}{|S|} \sum_{(x_n, y_n) \in S} \ell(y_n, f(x_n))$ for *loss function* $\ell \colon [L] \times \mathbb{R}^L \to \mathbb{R}_+$.

The standard classification setting requires making a prediction for *all* test samples. However, as we now detail, it is often prudent to allow the classsifer to *abstain* from predicting on some samples.

**Selective classification (SC)**. In *selective classification (SC)* (Geifman and El-Yaniv, 2019), closely related to the *learning to reject* or *learning with abstention* (Bartlett and Wegkamp, 2008; Cortes

et al., 2016a; Gangrade et al., 2021) problem, one may *abstain* from predicting on samples where a classifier has low-confidence. Intuitively, this allows for abstention on "hard" (e.g., ambiguously labelled) samples, which could be forwarded to an expert (e.g., a human labeller). Formally, given a budget $b_{rej} \in (0, 1)$ on the fraction of samples that can be rejected, one learns a classifier $h \colon \mathcal{X} \to \mathcal{Y}$ and *rejector* $r \colon \mathcal{X} \to \{0, 1\}$ to minimise the misclassification error on non-rejected samples:

$$\min_{h,r} \mathbb{P}_{in}(y \neq h(x), r(x) = 0) \colon \mathbb{P}_{in}(r(x) = 1) \leq b_{rej}. \tag{1}$$

The original SC formulation in Geifman and El-Yaniv (2019) conditions the misclassification error on samples that are *not* rejected; as shown in Appendix B, both formulations share the same optimal solution. The simplest SC baseline is *confidence-based* rejection (Chow, 1970; Ni et al., 2019), wherein $r$ thresholds the maximum of the *softmax probability* $p_y(x) \propto \exp(f_y(x))$. Alternatively, one may modify the training loss $\ell$ (Bartlett and Wegkamp, 2008; Ramaswamy et al., 2018; Charoenphakdee et al., 2021; Gangrade et al., 2021), or jointly learn an explicit rejector and classifier (Cortes et al., 2016a; Geifman and El-Yaniv, 2019; Thulasidasan et al., 2019; Mozannar and Sontag, 2020).

**OOD detection**. In *out-of-distribution* (*OOD*) *detection*, one seeks to identify test samples which are anomalous with respect to the training distribution (Hendrycks and Gimpel, 2017; Bendale and Boult, 2016; Bitterwolf et al., 2022). Intuitively, this allows one to abstain from predicting on samples where it is unreasonable to expect the classifier to generalise. This is closely related to the problem of detecting whether a sample is likely to be misclassified (Granese et al., 2021).

Formally, suppose $\mathbb{P}_{te} \doteq \pi_{in}^* \cdot \mathbb{P}_{in} + (1 - \pi_{in}^*) \cdot \mathbb{P}_{out}$, for (unknown) distribution $\mathbb{P}_{out}$ and $\pi_{in}^* \in (0, 1)$. Samples from $\mathbb{P}_{out}$ may be regarded as *outliers* or *out-of-distribution* with respect to the inlier distribution (ID) $\mathbb{P}_{in}$. Given a budget $b_{fpr} \in (0, 1)$ on the false positive rate (i.e., the fraction of ID samples incorrectly predicted as OOD), the goal is to learn an *OOD detector* $r \colon \mathcal{X} \to \{0, 1\}$ via

$$\min_r \mathbb{P}_{out}(r(x) = 0) \colon \mathbb{P}_{in}(r(x) = 1) \leq b_{fpr}. \tag{2}$$

*Labelled OOD detection* (Lee et al., 2018; Thulasidasan et al., 2019) additionally accounts for the accuracy of $h$. OOD detection is a natural task in the real-world, as standard classifiers may produce high-confidence predictions even on completely arbitrary inputs (Nguyen et al., 2015; Hendrycks and Gimpel, 2017), and assign higher scores to OOD compared to ID samples (Nalisnick et al., 2019).

Analogous to SC, a remarkably effective baseline for OOD detection that requires only ID samples is the *maximum softmax probability* (Hendrycks and Gimpel, 2017), possibly with temperature scaling and data augmentation (Liang et al., 2018). Recent works found that the maximum *logit* (Vaze et al., 2021; Hendrycks et al., 2022; Wei et al., 2022), and energy-based variants (Liu et al., 2020b) may be preferable. These may be further improved by taking into account imbalance in the ID classes (Jiang et al., 2023) and employing watermarking strategies (Wang et al., 2022b). More effective detectors can be designed in settings where one additionally has access to an OOD sample (Hendrycks et al., 2019; Thulasidasan et al., 2019; Dhamija et al., 2018; Katz-Samuels et al., 2022).

**Selective classification with OOD detection (SCOD)**. SC and OOD detection both involve abstaining from prediction, but for subtly different reasons: SC concerns *in-distribution but difficult* samples, while OOD detection concerns *out-of-distribution* samples. In practical classifier deployment, one is likely to encounter both types of samples. To this end, *selective classification with OOD detection* (SCOD) (Kim et al., 2021; Xia and Bouganis, 2022) allows for abstention on each sample type, with a user-specified parameter controlling their relative importance. Formally, suppose as before that $\mathbb{P}_{te} = \pi_{in}^* \cdot \mathbb{P}_{in} + (1 - \pi_{in}^*) \cdot \mathbb{P}_{out}$. Given a budget $b_{rej} \in (0, 1)$ on the fraction of test samples that can be rejected, the goal is to learn a classifier $h \colon \mathcal{X} \to \mathcal{Y}$ and a rejector $r \colon \mathcal{X} \to \{0, 1\}$ to minimise:

$$\min_{h,r} (1 - c_{fn}) \cdot \mathbb{P}_{in}(y \neq h(x), r(x) = 0) + c_{fn} \cdot \mathbb{P}_{out}(r(x) = 0) \colon \mathbb{P}_{te}(r(x) = 1) \leq b_{rej}. \tag{3}$$

Here, $c_{fn} \in [0, 1]$ is a user-specified cost of not rejecting an OOD sample. In Appendix C, we discuss alternate formulations for SCOD, and explain how our results seamlessly extend to such variants.

**Contrasting SCOD, SC, and OOD detection.** Before proceeding, it is worth pausing to emphasise the distinction between the three problems introduced above. All problems involve learning a rejector to enable the classifier from abstaining on certain samples. Crucially, SCOD encourages rejection on both ID samples that are likely to be misclassified, *and* OOD samples; by contrast, the SC and OOD detection problems only focus on one of these cases. Recent work has observed that standard OOD detectors tend to reject misclassified ID samples (Cen et al., 2023); thus, not considering the latter can lead to overly pessimistic estimates of rejector performance.

Given the practical relevance of SCOD, it is of interest to design effective techniques for the problem, analogous to those for SC and OOD detection. Surprisingly, the literature offers only a few instances of such techniques, most notably the SIRC method of Xia and Bouganis (2022). While empirically effective, this approach is heuristic in nature. We seek to design theoretically grounded techniques that are equally effective. To that end, we begin by investigating a fundamental property of SCOD.

Concurrent to this paper, we became aware of the highly related work of Franc et al. (2023), who provide optimality characterisations for SCOD-like formulations. In another concurrent work, Chaudhuri and Lopez-Paz (2023) seek to jointly calibrate a model for both SC and OOD detection.

## 3 BAYES-OPTIMAL SELECTIVE CLASSIFICATION WITH OOD DETECTION

We begin our formal analysis of SCOD by deriving its associated *Bayes-optimal* solution, which we show combines confidence scores for SC and density ratio scores for OOD detection.

### 3.1 BAYES-OPTIMAL SCOD RULE: SAMPLE-DEPENDENT CONFIDENCE THRESHOLDING

Before designing new techniques for SCOD, it is prudent to ask: what are the theoretically optimal choices for $h, r$ that we hope to approximate? More precisely, we seek to explicate the population SCOD objective (3) minimisers over *all* possible classifiers $h\colon \mathcal{X} \to \mathcal{Y}$, and rejectors $r\colon \mathcal{X} \to \{0,1\}$. These minimisers will depend on the unknown distributions $\mathbb{P}_{\text{in}}, \mathbb{P}_{\text{te}}$, and are thus not practically realisable as-is; nonetheless, they will subsequently motivate the design of simple, effective, and theoretically grounded solutions to SCOD. Further, these help study the efficacy of existing baselines.

Under mild distributional assumptions, one can apply a standard Lagrangian analysis (detailed in Appendix D) to show that (3) is equivalent to minimising over $h, r$:

$$L_{\text{scod}}(h, r)$$
$$= (1 - c_{\text{in}} - c_{\text{out}}) \cdot \mathbb{P}_{\text{in}}(y \neq h(x), r(x) = 0) + c_{\text{in}} \cdot \mathbb{P}_{\text{in}}(r(x) = 1) + c_{\text{out}} \cdot \mathbb{P}_{\text{out}}(r(x) = 0). \quad (4)$$

Here, $c_{\text{in}}, c_{\text{out}} \in [0, 1]$ are distribution-dependent constants which encode the false negative outlier cost $c_{\text{fn}}$, abstention budget $b_{\text{rej}}$, and the proportion $\pi_{\text{in}}^*$ of inliers in $\mathbb{P}_{\text{te}}$. We shall momentarily treat these constants as fixed and known; we return to the issue of suitable choices for them in §4.3. Furthermore, this formulation is fairly general and can be used to capture a variety of alternate constraints in (3) for specific choices of $c_{\text{in}}$ and $c_{\text{out}}$ (details in Appendix C). Note that we obtain a soft-penalty version of the SC problem when $c_{\text{out}} = 0$, and the OOD detection problem when $c_{\text{in}} + c_{\text{out}} = 1$. In general, we have the following Bayes-optimal solution for (4).

**Lemma 3.1.** *Let $(h^*, r^*)$ denote any minimiser of (3). Then, for any $x \in \mathcal{X}$ with $\mathbb{P}_{\text{in}}(x) > 0$:*

$$r^*(x) = \mathbf{1}\Big( (1 - c_{\text{in}} - c_{\text{out}}) \cdot \Big( 1 - \max_{y \in [L]} \mathbb{P}_{\text{in}}(y \mid x) \Big) + c_{\text{out}} \cdot \tfrac{\mathbb{P}_{\text{out}}(x)}{\mathbb{P}_{\text{in}}(x)} > c_{\text{in}} \Big). \quad (5)$$

*Further, $r^*(x) = 1$ when $\mathbb{P}_{\text{in}}(x) = 0$, and $h^*(x) = \operatorname{argmax}_{y \in [L]} \mathbb{P}_{\text{in}}(y \mid x)$ when $r^*(x) = 0$.*

The optimal classifier $h^*$ has an unsurprising form: for non-rejected samples, we predict the label $y$ with highest inlier class-probability $\mathbb{P}_{\text{in}}(y \mid x)$. The Bayes-optimal rejector is more interesting, and involves a comparison between two key quantities: the *maximum inlier class-probability* $\max_{y \in [L]} \mathbb{P}_{\text{in}}(y \mid x)$, and the *density ratio* $\frac{\mathbb{P}_{\text{in}}(x)}{\mathbb{P}_{\text{out}}(x)}$. These respectively reflect the confidence in the most likely label, and the confidence in the sample being an inlier. Intuitively, when either of these quantities is sufficiently small, a sample is a candidate for rejection.

We now verify that Lemma 3.1 generalises existing Bayes-optimal rules for SC and OOD detection.

**Special case: SC**. Suppose $c_{\text{out}} = 0$ and $c_{\text{in}} < 1$. Then, (5) reduces to *Chow's rule* (Chow, 1970; Ramaswamy et al., 2018):

$$r^*(x) = 1 \iff 1 - \max_{y \in [L]} \mathbb{P}_{\text{in}}(y \mid x) > \tfrac{c_{\text{in}}}{1 - c_{\text{in}}}. \quad (6)$$

Thus, samples with high uncertainty in the label distribution are rejected.

**Special case: OOD detection**. Suppose $c_{\text{in}} + c_{\text{out}} = 1$ and $c_{\text{in}} < 1$. Then, (5) reduces to *density-based rejection* (Steinwart et al., 2005; Chandola et al., 2009) when $\mathbb{P}_{\text{in}}(x) > 0$:

$$r^*(x) = 1 \iff \tfrac{\mathbb{P}_{\text{out}}(x)}{\mathbb{P}_{\text{in}}(x)} > \tfrac{c_{\text{in}}}{1 - c_{\text{in}}}. \quad (7)$$

Thus, samples with relatively high density under $\mathbb{P}_{\text{out}}$ are rejected.

## 3.2 IMPLICATION: EXISTING SC AND OOD BASELINES DO NOT SUFFICE FOR SCOD

Lemma 3.1 implies that SCOD cannot be readily solved by existing SC and OOD detection baselines. Specifically, consider the *confidence-based rejection* baseline, which rejects samples where $\max_{y \in [L]} \mathbb{P}_{\text{in}}(y \mid x)$ is lower than a fixed constant. This is known as *Chow's rule* (6) in the SC literature (Chow, 1970; Ramaswamy et al., 2018; Ni et al., 2019), and the *maximum softmax probability* (*MSP*) in OOD literature (Hendrycks and Gimpel, 2017); for brevity, we adopt the latter terminology. The MSP baseline does not suffice for the SCOD problem in general: even if $\max_{y \in [L]} \mathbb{P}_{\text{in}}(y \mid x) \sim 1$, it may be optimal to reject an input $x \in \mathcal{X}$ if $\mathbb{P}_{\text{out}}(x) \gg \mathbb{P}_{\text{in}}(x)$.

In fact, the MSP may result in *arbitrarily bad* rejection decisions. Surprisingly, this even holds in a special cases of OOD detection, such as *open-set classification*, wherein there is a strong relationship between $\mathbb{P}_{\text{in}}$ and $\mathbb{P}_{\text{out}}$ that *a-priori* would appear favourable to the MSP (Scheirer et al., 2013; Vaze et al., 2021). We elaborate on this with concrete examples in Appendix I.1.

One may ask whether using the maximum *logit* rather than softmax probability can prove successful in the open-set setting. Unfortunately, as this similarly does not include information about $\mathbb{P}_{\text{out}}$, it can also fail. For the same reason, other baselines from the OOD and SC literature can also fail; see Appendix I.3. Rather than using existing baselines as-is, we now consider a more direct approach to estimating the Bayes-optimal SCOD rejector in (5), which has strong empirical performance.

## 4 PLUG-IN ESTIMATORS TO THE BAYES-OPTIMAL SCOD RULE

The Bayes-optimal rule in (5) provides a prescription for how to combine estimates of confidence scores $s_{\text{sc}}^*(x) \doteq \max_{y \in [L]} \mathbb{P}_{\text{in}}(y \mid x)$ and density ratios $s_{\text{ood}}^*(x) \doteq \frac{\mathbb{P}_{\text{in}}(x)}{\mathbb{P}_{\text{out}}(x)}$ to make optimal rejection decisions for SCOD. Of course, obtaining reliable estimates of both quantities can be challenging.

Our focus in this paper is *not* to offer new approaches for estimating either of these quantities; rather, we seek to leverage existing selective classification and OOD detection techniques to estimate $s_{\text{sc}}^*(x)$ and $s_{\text{ood}}^*(x)$, and demonstrate how *optimally combining* the two scores leads to improved SCOD performance in practice. We also show theoretically that the efficacy of the resulting solution would indeed depend on the quality of the individual estimates (Lemmas 4.1 and 4.2), as is also the case with prior SCOD approaches (Xia and Bouganis, 2022).

To this end, we show how this combination strategy can be applied to two popular settings in the OOD detection literature: one where there is access to only samples from $\mathbb{P}_{\text{in}}$, and the other where there is also access to an unlabeled mix of ID and OOD samples (Katz-Samuels et al., 2022).

### 4.1 BLACK-BOX SCOD USING ONLY ID DATA

The first setting we consider assumes access to ID samples from $\mathbb{P}_{\text{in}}$. One may use *any* existing SC score — e.g., the maximum softmax probability estimate of Chow (1970) — to obtain estimates $s_{\text{sc}} \colon \mathcal{X} \to \mathbb{R}$ of the SC score. Similarly, we can leverage *any* existing OOD detection score $s_{\text{ood}} \colon \mathcal{X} \to \mathbb{R}$ that is computed only from ID data, e.g., the gradient norm score of Huang et al. (2021). Given these scores, we propose the following *black-box rejector*:

$$r_{\text{BB}}(x) = \mathbf{1}\left((1 - c_{\text{in}} - c_{\text{out}}) \cdot s_{\text{sc}}(x) + c_{\text{out}} \cdot \vartheta\left(s_{\text{ood}}(x)\right) < t_{\text{BB}}\right), \tag{8}$$

where $t_{\text{BB}} \doteq 1 - 2 \cdot c_{\text{in}} - c_{\text{out}}$, and $\vartheta \colon z \mapsto -\frac{1}{z}$. Observe that (8) exactly coincides with the optimal rejector (5) when $s_{\text{sc}}, s_{\text{ood}}$ equal their optimal counterparts $s_{\text{sc}}^*(x) \doteq \max_{y \in [L]} \mathbb{P}_{\text{in}}(y \mid x)$ and $s_{\text{ood}}^*(x) \doteq \frac{\mathbb{P}_{\text{in}}(x)}{\mathbb{P}_{\text{out}}(x)}$. Thus, as is intuitive, $r_{\text{BB}}$ will perform well when $s_{\text{sc}}, s_{\text{ood}}$ perform well on their respective tasks. Below, we bound the excess risk for $r_{\text{BB}}$ in terms errors in the estimated scores (which can be further bounded if, e.g., the scores are a result of minimising a surrogate loss).

**Lemma 4.1.** *Suppose we have estimates $\hat{\mathbb{P}}_{\text{in}}(y \mid x)$ of the inlier class probabilities $\mathbb{P}_{\text{in}}(y \mid x)$, estimates $\hat{s}_{\text{ood}}(x)$ of the density ratio $\frac{\mathbb{P}_{\text{in}}(x)}{\mathbb{P}_{\text{out}}(x)}$, and SC scores $\hat{s}_{\text{sc}}(x) = \max_{y \in [L]} \hat{\mathbb{P}}_{\text{in}}(y \mid x)$. Let $\hat{h}(x) \in \arg\max_{y \in [L]} \hat{\mathbb{P}}_{\text{in}}(y \mid x)$, and $\hat{r}_{\text{BB}}$ be a rejector defined according to (8) from $\hat{s}_{\text{sc}}(x)$ and $\hat{s}_{\text{ood}}(x)$. Let $\mathbb{P}^*(x) = 0.5 \cdot (\mathbb{P}_{\text{in}}(x) + \mathbb{P}_{\text{out}}(x))$. Then, for the SCOD-risk (3) minimizers $(h^*, r^*)$:*

$$L_{\text{scod}}(\hat{h}, \hat{r}_{\text{BB}}) - L_{\text{scod}}(h^*, r^*) \leq 4 \mathbb{E}_{x \sim \mathbb{P}^*}\left[\sum_y \left|\mathbb{P}_{\text{in}}(y \mid x) - \hat{\mathbb{P}}_{\text{in}}(y \mid x)\right| + \left|\frac{\mathbb{P}_{\text{in}}(x)}{\mathbb{P}_{\text{in}}(x) + \mathbb{P}_{\text{out}}(x)} - \frac{\hat{s}_{\text{ood}}(x)}{1 + \hat{s}_{\text{ood}}(x)}\right|\right].$$

**Sub-optimality of SIRC method (Xia and Bouganis, 2022).** Interestingly, this black-box rejector can be seen as a principled variant of the SIRC method of Xia and Bouganis (2022). As with $r_{\mathrm{BB}}$, SIRC works by combining rejection scores $s_{\mathrm{sc}}(x), s_{\mathrm{ood}}(x)$ for SC and OOD detection respectively. The key difference is that SIRC employs a *multiplicative* combination:

$$r_{\mathrm{SIRC}}(x) = 1 \iff (s_{\mathrm{sc}}(x) - a_1) \cdot \varrho(a_2 \cdot s_{\mathrm{ood}}(x) + a_3) < t_{\mathrm{SIRC}}, \qquad (9)$$

for constants $a_1, a_2, a_3$, threshold $t_{\mathrm{SIRC}}$, and monotone transform $\varrho \colon z \mapsto 1 + e^{-z}$. Intuitively, one rejects samples where there is sufficient signal that the sample is both near the decision boundary, *and* likely drawn from the outlier distribution. While empirically effective, it is not hard to see that the Bayes-optimal rejector (5) does not take the form of (9); thus, in general, SIRC may be sub-optimal. We note that this also holds for the objective considered in Xia and Bouganis (2022), which is a slight variation of (3) that enforces a constraint on the ID recall.

## 4.2 Loss-based SCOD using ID and OOD data

The second setting we consider is that of Katz-Samuels et al. (2022), which assumes access to both ID data, *and* a "wild" unlabeled sample comprising a mixture of ID and OOD data. As noted by Katz-Samuels et al., unlabeled "wild" data is typically plentiful, and can be collected, for example, from a currently deployed machine learning production system.

In this setting, the literature offers different loss functions (Hendrycks et al., 2019; Thulasidasan et al., 2021; Bitterwolf et al., 2022) to jointly estimate both the SC and OOD scores. We pick an adaptation of the *decoupled* loss proposed in Bitterwolf et al. (2022) due to its simplicity. We first describe this loss, assuming access to "clean" samples from $\mathbb{P}_{\mathrm{out}}$ and then explain how this loss can be applied to more practical settings where we have access to only "wild" samples.

Specifically, we learn scorers $f \colon \mathcal{X} \to \mathbb{R}^L$ and $s \colon \mathcal{X} \to \mathbb{R}$, with the goal of applying a suitable transformation to $f_y(x)$ and $s(x)$ to approximate $\mathbb{P}_{\mathrm{in}}(y \mid x)$ and $\frac{\mathbb{P}_{\mathrm{in}}(x)}{\mathbb{P}_{\mathrm{out}}(x)}$. We propose to minimise:

$$\mathbb{E}_{(x,y)\sim\mathbb{P}_{\mathrm{in}}} \left[\ell_{\mathrm{mc}}(y, f(x))\right] + \mathbb{E}_{x\sim\mathbb{P}_{\mathrm{in}}} \left[\ell_{\mathrm{bc}}(+1, s(x))\right] + \mathbb{E}_{x\sim\mathbb{P}_{\mathrm{out}}} \left[\ell_{\mathrm{bc}}(-1, s(x))\right], \qquad (10)$$

where $\ell_{\mathrm{mc}} \colon [L] \times \mathbb{R}^L \to \mathbb{R}_+$ and $\ell_{\mathrm{bc}} \colon \{\pm 1\} \times \mathbb{R} \to \mathbb{R}_+$ are *strictly proper composite* (Reid and Williamson, 2010) losses for multi-class and binary classification respectively. Canonical instantiations are the softmax cross-entropy $\ell_{\mathrm{mc}}(y, f(x)) = \log\left[\sum_{y'\in[L]} e^{f_{y'}(x)}\right] - f_y(x)$, and the sigmoid cross-entropy $\ell_{\mathrm{bc}}(z, s(x)) = \log(1 + e^{-z \cdot s(x)})$. In words, we use a standard multi-class classification loss on the ID data, with an additional loss that discriminates between the ID and OOD data. Note that in the last two terms, we do *not* impose separate costs for the OOD detection errors.

**Lemma 4.2.** *Let $\mathbb{P}^*(x, z) = \frac{1}{2} \left(\mathbb{P}_{\mathrm{in}}(x) \cdot \mathbf{1}(z = 1) + \mathbb{P}_{\mathrm{out}}(x) \cdot \mathbf{1}(z = -1)\right)$ denote a joint ID-OOD distribution, with $z = -1$ indicating an OOD sample. Suppose $\ell_{\mathrm{mc}}, \ell_{\mathrm{bc}}$ correspond to the softmax and sigmoid cross-entropy. Let $(f^*, s^*)$ be the minimizer of the decoupled loss in (10). For any scorers $f, s$, with transformations $p_y(x) = \frac{\exp(f_y(x))}{\sum_{y'} \exp(f_{y'}(x))}$ and $p_\perp(x) = \frac{1}{1+\exp(-s(x))}$:*

$$\mathbb{E}_{x\sim\mathbb{P}_{\mathrm{in}}} \left[\sum_{y\in[L]} \left|p_y(x) - \mathbb{P}_{\mathrm{in}}(y \mid x)\right|\right] \le \sqrt{2}\sqrt{\mathbb{E}_{(x,y)\sim\mathbb{P}_{\mathrm{in}}} \left[\ell_{\mathrm{mc}}(y, f(x))\right] - \mathbb{E}_{(x,y)\sim\mathbb{P}_{\mathrm{in}}} \left[\ell_{\mathrm{mc}}(y, f^*(x))\right]}$$

$$\mathbb{E}_{x\sim\mathbb{P}^*} \left[\left|p_\perp(x) - \frac{\mathbb{P}_{\mathrm{in}}(x)}{\mathbb{P}_{\mathrm{in}}(x)+\mathbb{P}_{\mathrm{out}}(x)}\right|\right] \le \frac{1}{\sqrt{2}}\sqrt{\mathbb{E}_{(x,z)\sim\mathbb{P}^*} \left[\ell_{\mathrm{bc}}(z, s(x))\right] - \mathbb{E}_{(x,z)\sim\mathbb{P}^*} \left[\ell_{\mathrm{bc}}(z, s^*(x))\right]}.$$

See Appendix E for a detailed *generalization bound*. Thus the quality of the estimates $p_y(x)$ and $p_\perp(x)$ depend on how well we are able to optimize the classifcation loss $\ell_{\mathrm{mc}}$ and the rejector loss $\ell_{\mathrm{bc}}$ in (10). Note that $\ell_{\mathrm{mc}}$ uses only the classification scores $f_y(x)$, while $\ell_{\mathrm{bc}}$ uses only the rejector score $s(x)$. The two losses are thus *decoupled*. We may introduce coupling *implicitly*, by parameterising $f_{y'}(x) = w_{y'}^\top \Phi(x)$ and $s(x) = u^\top \Phi(x)$ for shared embedding $\Phi$; or *explicitly*, as follows.

**Practical algorithm: SCOD in the wild**. The loss in (10) requires estimating expectations under $\mathbb{P}_{\mathrm{out}}$. While obtaining access to a sample drawn from $\mathbb{P}_{\mathrm{out}}$ may be challenging, we adopt a similar strategy to Katz-Samuels et al. (2022), and assume access to two sets of *unlabelled* samples:

**(A1)** $S_{\mathrm{mix}}$, consisting of a mixture of inlier and outlier samples drawn i.i.d. from a mixture $\mathbb{P}_{\mathrm{mix}} = \pi_{\mathrm{mix}} \cdot \mathbb{P}_{\mathrm{in}} + (1 - \pi_{\mathrm{mix}}) \cdot \mathbb{P}_{\mathrm{out}}$ of samples observed in the *wild* (e.g., during deployment)

**(A2)** $S_{\mathrm{in}}^*$, consisting of samples certified to be *strictly inlier*, i.e., with $\mathbb{P}_{\mathrm{out}}(x) = 0, \forall x \in S_{\mathrm{in}}^*$

---

**Algorithm 1** Loss-based SCOD using an unlabeled mixture of ID and OOD data

---

1: **Input:** Labeled $S_{\text{in}} \sim \mathbb{P}_{\text{in}}$, Unlabeled $S_{\text{mix}} \sim \mathbb{P}_{\text{mix}}$, Strictly inlier $S^*_{\text{in}}$ with $\mathbb{P}_{\text{out}}(x) = 0$

2: **Parameters:** Costs $c_{\text{in}}, c_{\text{out}}$ (derived from $c_{\text{fn}}$ and $b_{\text{rej}}$ specified in (3))

3: **Surrogate loss:** Find minimizers $\hat{f} : \mathcal{X} \to \mathbb{R}^L$ and $\hat{s} : \mathcal{X} \to \mathbb{R}$ of the decoupled loss:

$$\frac{1}{|S_{\text{in}}|} \sum_{(x,y) \in S_{\text{in}}} \ell_{\text{mc}}(y, f(x)) + \frac{1}{|S_{\text{in}}|} \sum_{(x,y) \in S_{\text{in}}} \ell_{\text{bc}}(+1, s(x)) + \frac{1}{|S_{\text{mix}}|} \sum_{x \in S_{\text{mix}}} \ell_{\text{bc}}(-1, s(x))$$

4: **Inlier class probabilities:** $\hat{\mathbb{P}}_{\text{in}}(y|x) \doteq \frac{1}{Z} \cdot \exp(\hat{f}_y(x))$, where $Z = \sum_{y'} \exp(\hat{f}_{y'}(x))$

5: **Mixture proportion:** $\hat{\pi}_{\text{mix}} \doteq \frac{1}{|S^*_{\text{in}}|} \sum_{x \in S^*_{\text{in}}} \exp(-\hat{s}(x))$

6: **Density ratio:** $\hat{s}_{\text{ood}}(x) \doteq \left( \frac{1}{1 - \hat{\pi}_{\text{mix}}} \cdot (\exp(-\hat{s}(x)) - \hat{\pi}_{\text{mix}}) \right)^{-1}$

7: **Plug-in:** Plug estimates $\hat{\mathbb{P}}_{\text{in}}(y|x)$, $\hat{s}_{\text{ood}}(x)$, and costs $c_{\text{in}}, c_{\text{out}}$ into (8), and construct $(\hat{h}, \hat{r})$

8: **Output:** $\hat{h}, \hat{r}$

---

Assumption **(A1)** was employed in Katz-Samuels et al. (2022), and may be implemented by collecting samples encountered "in the wild" during deployment of the SCOD classifier and rejector. Assumption **(A2)** merely requires identifying samples that are clearly *not* OOD, and is not difficult to satisfy: it may be implemented in practice by either identifying prototypical training samples[1], or by simply selecting a random subset of the training sample. We follow the latter in our experiments.

Equipped with $S_{\text{mix}}$, following Katz-Samuels et al. (2022), we propose to use it to approximate expectations under $\mathbb{P}_{\text{out}}$. One challenge is that the rejection logit will now estimate $\frac{\mathbb{P}_{\text{in}}(x)}{\mathbb{P}_{\text{mix}}(x)}$, rather than $\frac{\mathbb{P}_{\text{in}}(x)}{\mathbb{P}_{\text{out}}(x)}$. To resolve this, it is not hard to show that by **(A2)**, one can estimate the latter via a simple transformation (see Appendix G). Plugging these estimates into (8) then gives us an approximation to the Bayes-optimal solution. We summarise this procedure in Algorithm 1 for the decoupled loss.

In Appendix F, we additionally discuss a "coupled" variant of the loss in (10), and explain how the proposed losses relate to existing losses for OOD detection.

### 4.3 CHOOSING $c_{\text{in}}$ AND $c_{\text{out}}$

So far, we have focused on minimising (4), which requires specifying $c_{\text{in}}, c_{\text{out}}$. These costs need to be chosen based on parameters specified in the primal SCOD formulation in (3), namely: the abstention budget $b_{\text{rej}}$, and the cost $c_{\text{fn}}$ for non-rejection of OOD samples. The latter is a trade-off parameter also required in Xia and Bouganis (2022), and indicates how risk-averse a user is to making predictions on OOD samples (a value close to 1 indicates almost no tolerance for predictions on OOD samples). Using the Lagrangian for (3), one may set $c'_{\text{out}} = c_{\text{fn}} - \lambda \cdot (1 - \pi^*_{\text{in}})$ and $c'_{\text{in}} = \lambda \cdot \pi^*_{\text{in}}$, where $\lambda$ is the Lagrange multiplier and $\pi^*_{\text{in}}$ is the proportion of ID samples in the test population[2]; the resulting rejector takes the form $r^*(x) = \mathbf{1}\left( (1 - c_{\text{fn}}) \cdot \left(1 - \max_{y \in [L]} \mathbb{P}_{\text{in}}(y \mid x)\right) + c'_{\text{out}} \cdot \frac{\mathbb{P}_{\text{out}}(x)}{\mathbb{P}_{\text{in}}(x)} > c'_{\text{in}} \right)$. We prescribe treating $\lambda$ as the *lone* tuning parameter, and tuning it so that the resulting rejector in (5) satisfies the budget constraint specified by $b_{\text{rej}}$. See Appendix H for further details.

## 5 EXPERIMENTAL RESULTS

We demonstrate the efficacy of our proposed plug-in approaches to SCOD on a range of image classification benchmarks from the OOD detection and SCOD literature (Bitterwolf et al., 2022; Katz-Samuels et al., 2022; Xia and Bouganis, 2022). We report results with both pre-trained models and models trained from scratch; the latter are averaged over *5 random trials*.

**Datasets.** We use CIFAR-100 (Krizhevsky, 2009) and ImageNet (Deng et al., 2009) as the in-distribution (ID) datasets, and SVHN (Netzer et al., 2011), Places365 (Zhou et al., 2017), LSUN (Yu et al., 2015) (original and resized), Texture (Cimpoi et al., 2014), CelebA (Liu et al., 2015), 300K Random Images (Hendrycks et al., 2019), OpenImages (Krasin et al., 2017), OpenImages-O (Wang

---

[1]As a practical example, if we were to classify images as either cats or dogs, it is not hard to collect images that clearly show either a cat or a dog, and these would constitute strictly ID samples.

[2]In industry production settings, one may be able to estimate $\pi^*_{\text{in}}$ through inspection of historical logged data.

Table 2: Area Under the Risk-Coverage Curve (AUC-RC) for methods trained with CIFAR-100 as the ID sample and a mix of CIFAR-100 and either 300K Random Images or Open Images as the wild sample ($c_{\text{fn}} = 0.75$). The wild set contains 10% ID and 90% OOD. Base model is ResNet-56. A * against a method indicates that it uses both ID and OOD samples for training. The test set contains 50% ID and 50% OOD samples. *Lower* is *better*.

| Method / $\mathbb{P}_{\text{out}}^{\text{te}}$ | ID + OOD training with $\mathbb{P}_{\text{out}}^{\text{tr}}$ = Random300K | | | | | ID + OOD training with $\mathbb{P}_{\text{out}}^{\text{tr}}$ = OpenImages | | | | |
|---|---|---|---|---|---|---|---|---|---|---|
| | SVHN | Places | LSUN | LSUN-R | Texture | SVHN | Places | LSUN | LSUN-R | Texture |
| MSP | 0.307 | 0.338 | 0.323 | 0.388 | 0.344 | 0.307 | 0.338 | 0.323 | 0.388 | 0.344 |
| MaxLogit | 0.281 | 0.327 | 0.302 | 0.368 | 0.332 | 0.281 | 0.327 | 0.302 | 0.368 | 0.332 |
| Energy | 0.282 | 0.328 | 0.302 | 0.370 | 0.327 | 0.282 | 0.328 | 0.302 | 0.370 | 0.327 |
| DOCTOR | 0.306 | 0.336 | 0.322 | 0.384 | 0.341 | 0.306 | 0.336 | 0.322 | 0.384 | 0.341 |
| SIRC [$L_1$] | 0.279 | 0.334 | 0.302 | 0.385 | 0.316 | 0.279 | 0.334 | 0.302 | 0.385 | 0.316 |
| SIRC [Res] | 0.258 | 0.333 | 0.289 | 0.383 | 0.311 | 0.258 | 0.333 | 0.289 | 0.383 | 0.311 |
| CCE* | 0.287 | 0.314 | 0.254 | 0.212 | 0.257 | 0.303 | 0.209 | 0.246 | 0.210 | 0.277 |
| DCE* | 0.294 | 0.325 | 0.246 | 0.211 | 0.258 | 0.352 | 0.213 | 0.263 | 0.214 | 0.292 |
| OE* | 0.312 | **0.305** | 0.260 | 0.204 | 0.259 | 0.318 | 0.202 | 0.259 | 0.204 | 0.297 |
| Plug-in BB [$L_1$] | 0.223 | 0.318 | **0.240** | 0.349 | **0.245** | 0.223 | 0.318 | **0.240** | 0.349 | **0.245** |
| Plug-in BB [Res] | **0.205** | 0.324 | **0.240** | 0.321 | 0.264 | **0.205** | 0.324 | **0.240** | 0.321 | 0.264 |
| Plug-in LB* | 0.289 | **0.305** | 0.243 | **0.187** | 0.249 | 0.315 | **0.182** | 0.267 | **0.186** | 0.292 |

et al., 2022a), iNaturalist-O (Huang and Li, 2021) and Colorectal (Kather et al., 2016) as the OOD datasets. For training, we use labeled ID samples and (optionally) an unlabeled "wild" mixture of ID and OOD samples ($\mathbb{P}_{\text{mix}} = \pi_{\text{mix}} \cdot \mathbb{P}_{\text{in}} + (1 - \pi_{\text{mix}}) \cdot \mathbb{P}_{\text{out}}^{\text{te}}$). For testing, we use OOD samples ($\mathbb{P}_{\text{out}}^{\text{te}}$) that may be different from those used in training ($\mathbb{P}_{\text{out}}^{\text{tr}}$). We train a ResNet-56 on CIFAR, and use a pre-trained BiT ResNet-101 on ImageNet (hyper-parameter details in Appendix J.1).

In experiments where we use both ID and OOD samples for training, the training set comprises of equal number of ID samples and wild samples. We hold out 5% of the original ID test set and use it as the "strictly inlier" sample needed to estimate $\pi_{\text{mix}}$ for Algorithm 1. Our final test set contains equal proportions of ID and OOD samples; we report results with other choices in Appendix J.5.

**Evaluation metrics.** Recall that our goal is to solve the constrained SCOD objective in (3). One way to measure performance with respect to this objective is to measure the area under the risk-coverage curve (AUC-RC), as considered in prior work (Kim et al., 2021; Xia and Bouganis, 2022). Concretely, we plot the joint risk in (3) as a function of samples abstained, and evaluate the area under the curve. This summarizes the rejector performance on both selective classification and OOD detection. For a fixed fraction $\hat{b}_{\text{rej}} = \frac{1}{|S_{\text{all}}|} \sum_{x \in S_{\text{all}}} \mathbf{1}(r(x) = 1)$ of abstained samples, we measure the joint risk as:

$$\frac{1}{Z} \left( (1 - c_{\text{fn}}) \cdot \sum_{(x,y) \in S_{\text{in}}} \mathbf{1}(y \neq h(x), r(x) = 0) + c_{\text{fn}} \cdot \sum_{x \in S_{\text{out}}} \mathbf{1}(r(x) = 0) \right), \quad (11)$$

where $Z = \sum_{x \in S_{\text{all}}} \mathbf{1}(r(x) = 0)$ conditions the risk on non-rejected samples, and $S_{\text{all}} = \{x : (x, y) \in S_{\text{in}}\} \cup S_{\text{out}}$ is the combined ID-OOD dataset. See Appendix H for details of how our plug-in estimators handle this constrained objective. We set $c_{\text{fn}} = 0.75$ here, and explore other cost parameters in Appendix J.6.

**Baselines.** Our *main competitor is SIRC* from the SCOD literature (Xia and Bouganis, 2022). We compare with two variants of SIRC, which respectively use the $L_1$-norm of the embeddings for $s_{\text{ood}}$, and a residual score (Wang et al., 2022a) instead. For completeness, we also include representative baselines from the OOD literature to show that a stand-alone OOD scorer can be sub-optimal for SCOD. We do not include an exhaustive list of OOD methods, as the task at hand is SCOD, and *not* stand-alone OOD detection. We include both methods that train only on the ID samples, namely, MSP (Chow, 1970; Hendrycks and Gimpel, 2017), MaxLogit (Hendrickx et al., 2021), energy-based scorer (Liu et al., 2020b), DOCTOR (Granese et al., 2021), and those which additionally use OOD samples, namely, the coupled CE loss (CCE) (Thulasidasan et al., 2021), the de-coupled CE loss (DCE) (Bitterwolf et al., 2022), and the outlier exposure (OE) (Hendrycks et al., 2019). In Appendix J, we also compare against cost-sensitive softmax (CSS) loss (Mozannar and Sontag, 2020), a representative SC baseline, nearest-neighbor scorers (Sun et al., 2022), and ODIN (Liang et al., 2018).

**Plug-in estimators.** For a fair comparison, we implement our *black-box* rejector in (8) using the *same $s_{\text{ood}}$ scorers* as Xia and Bouganis (2022), namely their (i) $L_1$ scorer and (ii) residual scorer; we

Table 3: AUC-RC ($\downarrow$) for CIFAR-100 as ID, and a "wild" comprising of 90% ID and *only* 10% OOD. The OOD part of the wild set is drawn from the *same* OOD dataset from which the test set is drawn.

| Method / $\mathbb{P}_{\text{out}}^{\text{te}}$ | SVHN | Places | LSUN | LSUN-R | Texture | OpenImages |
|---|---|---|---|---|---|---|
| | \multicolumn ID + OOD training with $\mathbb{P}_{\text{out}}^{\text{tr}} = \mathbb{P}_{\text{out}}^{\text{te}}$ | | | | | |
| MSP | 0.307 | 0.338 | 0.323 | 0.388 | 0.344 | 0.342 |
| MaxLogit | 0.281 | 0.327 | 0.302 | 0.368 | 0.332 | 0.351 |
| Energy | 0.282 | 0.328 | 0.302 | 0.370 | 0.327 | 0.351 |
| DOCTOR | 0.306 | 0.336 | 0.322 | 0.384 | 0.341 | 0.342 |
| SIRC [$L_1$] | 0.279 | 0.334 | 0.302 | 0.385 | 0.316 | 0.340 |
| SIRC [Res] | 0.258 | 0.333 | 0.289 | 0.383 | 0.311 | 0.341 |
| CCE* | 0.238 | 0.227 | 0.231 | 0.235 | 0.239 | 0.243 |
| DCE* | 0.235 | 0.220 | 0.226 | 0.230 | 0.235 | 0.241 |
| OE* | 0.245 | 0.245 | 0.254 | 0.241 | 0.264 | 0.255 |
| Plug-in BB [$L_1$] | 0.223 | 0.318 | 0.240 | 0.349 | 0.245 | 0.334 |
| Plug-in BB [Res] | **0.205** | 0.324 | 0.240 | 0.321 | 0.264 | 0.342 |
| Plug-in LB* | 0.221 | **0.199** | **0.209** | **0.215** | **0.218** | **0.225** |

Table 4: AUC-RC ($\downarrow$) for methods trained on ImageNet (inlier) with **no OOD samples**. The base model is a pre-trained BiT ResNet-101. *Lower* values are *better*. Additional results in App. J.9.

| Method / $\mathbb{P}_{\text{out}}^{\text{te}}$ | Places | LSUN | CelebA | Colorectal | iNaturalist-O | Texture | OpenImages-O | ImageNet-O |
|---|---|---|---|---|---|---|---|---|
| | \multicolumn ID-only training | | | | | | | |
| MSP | 0.227 | 0.234 | 0.241 | 0.218 | 0.195 | 0.220 | 0.203 | 0.325 |
| MaxLogit | 0.229 | 0.239 | 0.256 | 0.204 | 0.195 | 0.223 | **0.202** | 0.326 |
| Energy | 0.235 | 0.246 | 0.278 | 0.204 | 0.199 | 0.227 | 0.210 | 0.330 |
| DOCTOR | 0.220 | 0.233 | 0.235 | 0.220 | 0.193 | 0.226 | **0.202** | 0.331 |
| SIRC [$L_1$] | 0.222 | 0.229 | 0.248 | 0.220 | 0.196 | 0.226 | **0.200** | 0.313 |
| SIRC [Res] | 0.211 | 0.198 | 0.178 | 0.161 | 0.175 | 0.219 | **0.201** | 0.327 |
| Plug-in BB [$L_1$] | 0.261 | 0.257 | 0.337 | 0.283 | 0.219 | 0.270 | 0.222 | 0.333 |
| Plug-in BB [Res] | **0.191** | **0.170** | **0.145** | **0.149** | **0.162** | 0.252 | 0.215 | 0.378 |

use their MSP scorer for $s_{\text{sc}}$. We also include our (iii) *loss-based* rejector based on the de-coupled (DC) loss in (10). (i) and (ii) use only ID samples; (iii) uses both ID and wild samples for training.

**Tuning parameter.** Each baseline has a *single threshold* or cost parameter that needs to be tuned to achieve a given rate of abstention $b_{\text{rej}}$ (details in Appendix J.1); we aggregate performance across different abstention rates. Our plug-in method also uses a single tuning parameter (details in §4.3).

**Results.** Our first experiments use CIFAR-100 as the ID sample. Table 2 reports results for a setting where the OOD samples used (as a part of the wild set) during training are different from those used for testing ($\mathbb{P}_{\text{out}}^{\text{tr}} \neq \mathbb{P}_{\text{out}}^{\text{te}}$). Table 3 contains results for a setting where they are the same ($\mathbb{P}_{\text{out}}^{\text{tr}} = \mathbb{P}_{\text{out}}^{\text{te}}$). In both cases, *one among the three plug-in estimators yields the lowest AUC-RC*. Interestingly, when $\mathbb{P}_{\text{out}}^{\text{tr}} \neq \mathbb{P}_{\text{out}}^{\text{te}}$, the two black-box (BB) plug-in estimators that use only ID-samples for training often fare better than the loss-based (LB) one which uses both ID and wild samples for training. This is likely due to the mismatch between the training and test OOD distributions resulting in the decoupled loss yielding poor estimates of $\frac{\mathbb{P}_{\text{in}}(x)}{\mathbb{P}_{\text{out}}(x)}$. When $\mathbb{P}_{\text{out}}^{\text{tr}} = \mathbb{P}_{\text{out}}^{\text{te}}$, the LB estimator often performs the best.

Table 4 presents results with *ImageNet as ID*, and *no OOD samples for training*. The BB plug-in estimator (residual) yields notable gains on 5/8 OOD datasets. On the remaining, even the SIRC baselines are often only marginally better than MSP; this is because the grad-norm scorers used by them (and also by our estimators) are not very effective in detecting OOD samples for these datasets.

We have thus provided theoretically grounded plug-in estimators for SCOD that combine scores of SC and OOD detection, and demonstrated their efficacy on both settings that train with only ID samples, and those that additionally use a noisy OOD sample. A key element in our approach is the $s_{\text{ood}}$ scorer for estimating the ID-OOD density ratio, for which we employed the grad-norm based scorers (Wang et al., 2022a) used by prior SCOD methods (Xia and Bouganis, 2022). In the future, we wish to explore other approaches for estimating the density ratio such as Ren et al. (2019); Sun et al. (2022). It is also of interest to consider unifying the statistical formulation of SCOD with the problem of identifying misclassified samples (Hendrycks and Gimpel, 2017; Granese et al., 2021).

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

# Appendix

## A PROOFS

*Lemma 3.1.* Let $(h^*, r^*)$ denote any minimiser of (3). Then, for any $x \in \mathcal{X}$ with $\mathbb{P}_{\text{in}}(x) > 0$:

$$r^*(x) = \mathbf{1}\left(\left(1 - c_{\text{in}} - c_{\text{out}}\right) \cdot \left(1 - \max_{y \in [L]} \mathbb{P}_{\text{in}}(y \mid x)\right) + c_{\text{out}} \cdot \frac{\mathbb{P}_{\text{out}}(x)}{\mathbb{P}_{\text{in}}(x)} > c_{\text{in}}\right). \tag{12}$$

Further, $r^*(x) = 1$ when $\mathbb{P}_{\text{in}}(x) = 0$, and $h^*(x) = \operatorname{argmax}_{y \in [L]} \mathbb{P}_{\text{in}}(y \mid x)$ when $r^*(x) = 0$.

*Proof of Lemma 3.1.* We first define a joint marginal distribution $\mathbb{P}_{\text{comb}}$ that samples from $\mathbb{P}_{\text{in}}(x)$ and $\mathbb{P}_{\text{out}}(x)$ with equal probabilities. We then rewrite the objective in (4) in terms of the joint marginal distribution:

$$L_{\text{scod}}(h, r) = \mathbb{E}_{x \sim \mathbb{P}_{\text{comb}}}\left[T_1(h(x), r(x)) + T_2(h(x), r(x))\right]$$

$$T_1(h(x), r(x)) = (1 - c_{\text{in}} - c_{\text{out}}) \cdot \mathbb{E}_{y|x \sim \mathbb{P}_{\text{in}}}\left[\frac{\mathbb{P}_{\text{in}}(x)}{\mathbb{P}_{\text{comb}}(x)} \cdot \mathbf{1}(y \neq h(x), r(x) = 0)\right]$$

$$= (1 - c_{\text{in}} - c_{\text{out}}) \cdot \sum_{y \in [L]} \mathbb{P}_{\text{in}}(y|x) \cdot \frac{\mathbb{P}_{\text{in}}(x)}{\mathbb{P}_{\text{comb}}(x)} \cdot \mathbf{1}(y \neq h(x), r(x) = 0)$$

$$T_2(h(x), r(x)) = c_{\text{in}} \cdot \frac{\mathbb{P}_{\text{in}}(x)}{\mathbb{P}_{\text{comb}}(x)} \cdot \mathbf{1}(r(x) = 1) + c_{\text{out}} \cdot \mathbf{1}(r(x) = 0).$$

The conditional risk that a classifier $h$ incurs when abstaining (i.e., predicting $r(x) = 1$) on a fixed instance $x$ is given by:

$$c_{\text{in}} \cdot \frac{\mathbb{P}_{\text{in}}(x)}{\mathbb{P}_{\text{comb}}(x)}.$$

The conditional risk associated with predicting a base class $y \in [L]$ on instance $x$ is given by:

$$(1 - c_{\text{in}} - c_{\text{out}}) \cdot \frac{\mathbb{P}_{\text{in}}(x)}{\mathbb{P}_{\text{comb}}(x)} \cdot (1 - \mathbb{P}_{\text{in}}(y|x)) + c_{\text{out}} \cdot \frac{\mathbb{P}_{\text{out}}(x)}{\mathbb{P}_{\text{comb}}(x)}$$

The Bayes-optimal classifier then predicts the label with the lowest conditional risk. When $\mathbb{P}_{\text{in}}(x) = 0$, this amounts to predicting abstain ($r(x) = 1$). When $\mathbb{P}_{\text{in}}(x) > 0$, the optimal classifier predicts $r(x) = 1$ when:

$$c_{\text{in}} \cdot \frac{\mathbb{P}_{\text{in}}(x)}{\mathbb{P}_{\text{comb}}(x)} < (1 - c_{\text{in}} - c_{\text{out}}) \cdot \frac{\mathbb{P}_{\text{in}}(x)}{\mathbb{P}_{\text{comb}}(x)} \cdot \min_{y \in [L]}\left(1 - \mathbb{P}_{\text{in}}(y|x)\right) + c_{\text{out}} \cdot \frac{\mathbb{P}_{\text{out}}(x)}{\mathbb{P}_{\text{comb}}(x)}$$

$$\iff c_{\text{in}} \cdot \mathbb{P}_{\text{in}}(x) < (1 - c_{\text{in}} - c_{\text{out}}) \cdot \mathbb{P}_{\text{in}}(x) \cdot \min_{y \in [L]}\left(1 - \mathbb{P}_{\text{in}}(y|x)\right) + c_{\text{out}} \cdot \mathbb{P}_{\text{out}}(x)$$

$$\iff c_{\text{in}} \cdot \mathbb{P}_{\text{in}}(x) < (1 - c_{\text{in}} - c_{\text{out}}) \cdot \mathbb{P}_{\text{in}}(x) \cdot \left(1 - \max_{y \in [L]} \mathbb{P}_{\text{in}}(y|x)\right) + c_{\text{out}} \cdot \mathbb{P}_{\text{out}}(x)$$

$$\iff c_{\text{in}} < (1 - c_{\text{in}} - c_{\text{out}}) \cdot \left(1 - \max_{y \in [L]} \mathbb{P}_{\text{in}}(y|x)\right) + c_{\text{out}} \cdot \frac{\mathbb{P}_{\text{out}}(x)}{\mathbb{P}_{\text{in}}(x)}.$$

Otherwise, the classifier does not abstain ($r(x) = 0$), and predicts $\operatorname{argmax}_{y \in [L]} \mathbb{P}_{\text{in}}(y|x)$, as desired.
$\square$

*Lemma 4.1.* Suppose we have estimates $\hat{\mathbb{P}}_{\text{in}}(y \mid x)$ of the inlier class probabilities $\mathbb{P}_{\text{in}}(y \mid x)$, estimates $\hat{s}_{\text{ood}}(x)$ of the density ratio $\frac{\mathbb{P}_{\text{in}}(x)}{\mathbb{P}_{\text{out}}(x)}$, and *SC* scores $\hat{s}_{\text{sc}}(x) = \max_{y \in [L]} \hat{\mathbb{P}}_{\text{in}}(y \mid x)$. Let $\hat{h}(x) \in \operatorname{argmax}_{y \in [L]} \hat{\mathbb{P}}_{\text{in}}(y \mid x)$, and $\hat{r}_{\text{BB}}$ be a rejector defined according to (8) from $\hat{s}_{\text{sc}}(x)$ and $\hat{s}_{\text{ood}}(x)$. Let $\mathbb{P}^*(x) = \frac{1}{2}(\mathbb{P}_{\text{in}}(x) + \mathbb{P}_{\text{out}}(x))$. Then, for the *SCOD*-risk (3) minimizers $(h^*, r^*)$:

$$L_{\text{scod}}(\hat{h}, \hat{r}_{\text{BB}}) - L_{\text{scod}}(h^*, r^*)$$
$$\leq 2 \cdot \mathbb{E}_{x \sim \mathbb{P}^*}\left[\sum_{y \in [L]}\left|\mathbb{P}_{\text{in}}(y \mid x) - \hat{\mathbb{P}}_{\text{in}}(y \mid x)\right| + 4 \cdot \left|\frac{\mathbb{P}_{\text{in}}(x)}{\mathbb{P}_{\text{in}}(x) + \mathbb{P}_{\text{out}}(x)} - \frac{\hat{s}_{\text{ood}}(x)}{1 + \hat{s}_{\text{ood}}(x)}\right|\right].$$

*Proof of Lemma 4.1.* Let $\mathbb{P}^*$ denote the joint distribution that draws a sample from $\mathbb{P}_{\text{in}}$ and $\mathbb{P}_{\text{out}}$ with equal probability. Denote $\gamma_{\text{in}}(x) = \frac{\mathbb{P}_{\text{in}}(x)}{\mathbb{P}_{\text{in}}(x) + \mathbb{P}_{\text{out}}(x)}$. The joint risk in (4) can be written as:

$$
\begin{aligned}
&L_{\text{scod}}(h, r) \\
&= (1 - c_{\text{in}} - c_{\text{out}}) \cdot \mathbb{P}_{\text{in}}(y \neq h(x), r(x) = 0) + c_{\text{in}} \cdot \mathbb{P}_{\text{in}}(r(x) = 1) + c_{\text{out}} \cdot \mathbb{P}_{\text{out}}(r(x) = 0) \\
&= \mathbb{E}_{x \sim \mathbb{P}^*} \Big[ (1 - c_{\text{in}} - c_{\text{out}}) \cdot \gamma_{\text{in}}(x) \cdot \sum_{y \neq h(x)} \mathbb{P}_{\text{in}}(y \mid x) \cdot \mathbf{1}(r(x) = 0) \\
&\qquad\qquad + c_{\text{in}} \cdot \gamma_{\text{in}}(x) \cdot \mathbf{1}(r(x) = 1) + c_{\text{out}} \cdot (1 - \gamma_{\text{in}}(x)) \cdot \mathbf{1}(r(x) = 0) \Big].
\end{aligned}
$$

For class probability estimates $\hat{\mathbb{P}}_{\text{in}}(y \mid x) \approx \mathbb{P}_{\text{in}}(y \mid x)$, and scorers $\hat{s}_{\text{sc}}(x) = \max_{y \in [L]} \hat{\mathbb{P}}_{\text{in}}(y \mid x)$ and $\hat{s}_{\text{ood}}(x) \approx \frac{\mathbb{P}_{\text{in}}(x)}{\mathbb{P}_{\text{out}}(x)}$, we construct a classifier $\hat{h}(x) \in \text{argmax}_{y \in [L]} \hat{\eta}_y(x)$ and black-box rejector:

$$
\hat{r}_{\text{BB}}(x) = 1 \iff (1 - c_{\text{in}} - c_{\text{out}}) \cdot (1 - \hat{s}_{\text{sc}}(x)) + c_{\text{out}} \cdot \left( \frac{1}{\hat{s}_{\text{ood}}(x)} \right) > c_{\text{in}}. \tag{13}
$$

Let $(h^*, r^*)$ denote the optimal classifier and rejector as defined in (5). We then wish to bound the following regret:

$$
L_{\text{scod}}(\hat{h}, \hat{r}_{\text{BB}}) - L_{\text{scod}}(h^*, r^*) = \underbrace{L_{\text{scod}}(\hat{h}, \hat{r}_{\text{BB}}) - L_{\text{scod}}(h^*, \hat{r}_{\text{BB}})}_{\text{term}_1} + \underbrace{L_{\text{scod}}(h^*, \hat{r}_{\text{BB}}) - L_{\text{scod}}(h^*, r^*)}_{\text{term}_2}.
$$

We first bound the first term:

$$
\begin{aligned}
\text{term}_1 &= \mathbb{E}_{x \sim \mathbb{P}^*} \left[ (1 - c_{\text{in}} - c_{\text{out}}) \cdot \gamma_{\text{in}}(x) \cdot \mathbf{1}(\hat{r}_{\text{BB}}(x) = 0) \cdot \left( \sum_{y \neq \hat{h}(x)} \mathbb{P}_{\text{in}}(y \mid x) - \sum_{y \neq h^*(x)} \mathbb{P}_{\text{in}}(y \mid x) \right) \right] \\
&= \mathbb{E}_{x \sim \mathbb{P}^*} \left[ \omega(x) \cdot \left( \sum_{y \neq \hat{h}(x)} \mathbb{P}_{\text{in}}(y \mid x) - \sum_{y \neq h^*(x)} \mathbb{P}_{\text{in}}(y \mid x) \right) \right],
\end{aligned}
$$

where we denote $\omega(x) = (1 - c_{\text{in}} - c_{\text{out}}) \cdot \gamma_{\text{in}}(x) \cdot \mathbf{1}(\hat{r}_{\text{BB}}(x) = 0)$.

Furthermore, we can write:

$$
\text{term}_1
$$

$$
\begin{aligned}
&= \mathbb{E}_{x \sim \mathbb{P}^*} \left[ \omega(x) \cdot \left( \sum_{y \neq \hat{h}(x)} \mathbb{P}_{\text{in}}(y \mid x) - \sum_{y \neq h^*(x)} \hat{\mathbb{P}}_{\text{in}}(y \mid x) + \sum_{y \neq h^*(x)} \hat{\mathbb{P}}_{\text{in}}(y \mid x) - \sum_{y \neq h^*(x)} \mathbb{P}_{\text{in}}(y \mid x) \right) \right] \\
&\leq \mathbb{E}_{x \sim \mathbb{P}^*} \left[ \omega(x) \cdot \left( \sum_{y \neq \hat{h}(x)} \mathbb{P}_{\text{in}}(y \mid x) - \sum_{y \neq \hat{h}(x)} \hat{\mathbb{P}}_{\text{in}}(y \mid x) + \sum_{y \neq h^*(x)} \hat{\mathbb{P}}_{\text{in}}(y \mid x) - \sum_{y \neq h^*(x)} \mathbb{P}_{\text{in}}(y \mid x) \right) \right] \\
&\leq 2 \cdot \mathbb{E}_{x \sim \mathbb{P}^*} \left[ \omega(x) \cdot \sum_{y \in [L]} \left| \mathbb{P}_{\text{in}}(y \mid x) - \hat{\mathbb{P}}_{\text{in}}(y \mid x) \right| \right] \\
&\leq 2 \cdot \mathbb{E}_{x \sim \mathbb{P}^*} \left[ \sum_{y \in [L]} \left| \mathbb{P}_{\text{in}}(y \mid x) - \hat{\mathbb{P}}_{\text{in}}(y \mid x) \right| \right],
\end{aligned}
$$

where the third step uses the definition of $\hat{h}$ and the fact that $\omega(x) > 0$; the last step uses the fact that $\omega(x) \leq 1$.

We bound the second term now. For this, we first define:

$$
L_{\text{rej}}(r) = \mathbb{E}_{x \sim \mathbb{P}^*} \left[ \left( (1 - c_{\text{in}} - c_{\text{out}}) \cdot \gamma_{\text{in}}(x) \cdot (1 - \max_{y \in [L]} \mathbb{P}_{\text{in}}(y \mid x)) + c_{\text{out}} \cdot (1 - \gamma_{\text{in}}(x)) \right) \cdot \mathbf{1}(r(x) = 0)
$$

$$+ c_{\text{in}} \cdot \gamma_{\text{in}}(x) \cdot \mathbf{1}(r(x) = 1) \Bigg].$$

and

$$\hat{L}_{\text{rej}}(r) = \mathbb{E}_{x \sim \mathbb{P}^*} \Bigg[ \left( (1 - c_{\text{in}} - c_{\text{out}}) \cdot \hat{\gamma}_{\text{in}}(x) \cdot (1 - \max_{y \in [L]} \hat{\mathbb{P}}_{\text{in}}(y \mid x)) + c_{\text{out}} \cdot (1 - \hat{\gamma}_{\text{in}}(x)) \right) \cdot \mathbf{1}(r(x) = 0)$$

$$+ c_{\text{in}} \cdot \hat{\gamma}_{\text{in}}(x) \cdot \mathbf{1}(r(x) = 1) \Bigg],$$

where we denote $\hat{\gamma}_{\text{in}}(x) = \frac{\hat{s}_{\text{ood}}(x)}{1 + \hat{s}_{\text{ood}}(x)}$.

Notice that $r^*$ minimizes $L(r)$ over all rejectors $r : \mathcal{X} \to \{0, 1\}$. Similarly, note that $\hat{r}_{\text{BB}}$ minimizes $\hat{L}(r)$ over all rejectors $r : \mathcal{X} \to \{0, 1\}$.

Then the second term can be written as:

$$\text{term}_2 = L_{\text{rej}}(\hat{r}_{\text{BB}}) - L_{\text{rej}}(r^*)$$

$$= L_{\text{rej}}(\hat{r}_{\text{BB}}) - \hat{L}_{\text{rej}}(r^*) + \hat{L}_{\text{rej}}(r^*) - L_{\text{rej}}(r^*)$$

$$\leq L_{\text{rej}}(\hat{r}_{\text{BB}}) - \hat{L}_{\text{rej}}(\hat{r}_{\text{BB}}) + \hat{L}_{\text{rej}}(r^*) - L_{\text{rej}}(r^*)$$

$$\leq 2 \cdot (1 - c_{\text{in}} - c_{\text{out}}) \cdot \left| \max_{y \in [L]} \mathbb{P}_{\text{in}}(y \mid x) - \max_{y \in [L]} \hat{\mathbb{P}}_{\text{in}}(y \mid x) \right| \cdot |\gamma_{\text{in}}(x) - \hat{\gamma}_{\text{in}}(x)|$$

$$+ 2 \cdot \left( (1 - c_{\text{in}} - c_{\text{out}}) + c_{\text{out}} + c_{\text{in}} \right) \cdot |\gamma_{\text{in}}(x) - \hat{\gamma}_{\text{in}}(x)|$$

$$\leq 2 \cdot (1 - c_{\text{in}} - c_{\text{out}}) \cdot (1) \cdot |\gamma_{\text{in}}(x) - \hat{\gamma}_{\text{in}}(x)| + 2 \cdot (1) \cdot |\gamma_{\text{in}}(x) - \hat{\gamma}_{\text{in}}(x)|$$

$$\leq 4 \cdot |\gamma_{\text{in}}(x) - \hat{\gamma}_{\text{in}}(x)|$$

$$= 4 \cdot \left| \frac{\mathbb{P}_{\text{in}}(x)}{\mathbb{P}_{\text{in}}(x) + \mathbb{P}_{\text{out}}(x)} - \frac{\hat{s}_{\text{ood}}(x)}{1 + \hat{s}_{\text{ood}}(x)} \right|,$$

where the third step follows from $\hat{r}_{\text{BB}}$ being a minimizer of $\hat{L}_{\text{rej}}(r)$, the fourth step uses the fact that $\left| \max_{y \in [L]} \mathbb{P}_{\text{in}}(y \mid x) - \max_{y \in [L]} \hat{\mathbb{P}}_{\text{in}}(y \mid x) \right| \leq 1$, and the fifth step uses the fact that $c_{\text{in}} + c_{\text{out}} \leq 1$.

Combining the bounds on $\text{term}_1$ and $\text{term}_2$ completes the proof. $\qquad\square$

*Lemma 4.2.* Let $\mathbb{P}^*(x, z) = \frac{1}{2} \left( \mathbb{P}_{\text{in}}(x) \cdot \mathbf{1}(z = 1) + \mathbb{P}_{\text{out}}(x) \cdot \mathbf{1}(z = -1) \right)$ denote a joint ID-OOD distribution, with $z = -1$ indicating an OOD sample. Suppose $\ell_{\text{mc}}, \ell_{\text{bc}}$ correspond to the softmax and sigmoid cross-entropy. Let $(f^*, s^*)$ be the minimizer of the decoupled loss in (10). For any scorers $f, s$, with transformations $p_y(x) = \frac{\exp(f_y(x))}{\sum_{y'} \exp(f_{y'}(x))}$ and $p_\perp(x) = \frac{1}{1 + \exp(-s(x))}$:

$$\mathbb{E}_{x \sim \mathbb{P}_{\text{in}}} \left[ \sum_{y \in [L]} |p_y(x) - \mathbb{P}_{\text{in}}(y \mid x)| \right] \leq \sqrt{2} \sqrt{\mathbb{E}_{(x,y) \sim \mathbb{P}_{\text{in}}} [\ell_{\text{mc}}(y, f(x))] - \mathbb{E}_{(x,y) \sim \mathbb{P}_{\text{in}}} [\ell_{\text{mc}}(y, f^*(x))]}$$

$$\mathbb{E}_{x \sim \mathbb{P}^*} \left[ \left| p_\perp(x) - \frac{\mathbb{P}_{\text{in}}(x)}{\mathbb{P}_{\text{in}}(x) + \mathbb{P}_{\text{out}}(x)} \right| \right] \leq \frac{1}{\sqrt{2}} \sqrt{\mathbb{E}_{(x,z) \sim \mathbb{P}^*} [\ell_{\text{bc}}(z, s(x))] - \mathbb{E}_{(x,z) \sim \mathbb{P}^*} [\ell_{\text{bc}}(z, s^*(x))]}.$$

*Proof of Lemma 4.2.* We first note that $f^*(x) \propto \log(\mathbb{P}_{\text{in}}(y \mid x))$ and $s^*(x) = \log \left( \frac{\mathbb{P}^*(z=1|x)}{\mathbb{P}^*(z=-1|x)} \right)$.

**Regret Bound 1**: We start with the first regret bound. We expand the multi-class cross-entropy loss to get:

$$\mathbb{E}_{(x,y) \sim \mathbb{P}_{\text{in}}} [\ell_{\text{mc}}(y, f(x))] = \mathbb{E}_{x \sim \mathbb{P}_{\text{in}}} \left[ - \sum_{y \in [L]} \mathbb{P}_{\text{in}}(y \mid x) \cdot \log(p_y(x)) \right]$$

$$\mathbb{E}_{(x,y) \sim \mathbb{P}_{\text{in}}} [\ell_{\text{mc}}(y, f^*(x))] = \mathbb{E}_{x \sim \mathbb{P}_{\text{in}}} \left[ - \sum_{y \in [L]} \mathbb{P}_{\text{in}}(y \mid x) \cdot \log(\mathbb{P}_{\text{in}}(y \mid x)) \right].$$

The right-hand side of the first bound can then be expanded as:

$$\mathbb{E}_{(x,y) \sim \mathbb{P}_{\text{in}}} [\ell_{\text{mc}}(y, f(x))] - \mathbb{E}_{(x,y) \sim \mathbb{P}_{\text{in}}} [\ell_{\text{mc}}(y, f^*(x))] = \mathbb{E}_{x \sim \mathbb{P}_{\text{in}}} \left[ \sum_{y \in [L]} \mathbb{P}_{\text{in}}(y \mid x) \cdot \log \left( \frac{\mathbb{P}_{\text{in}}(y \mid x)}{p_y(x)} \right) \right], \tag{14}$$

which the KL-divergence between $\mathbb{P}_{\mathrm{in}}(y \mid x)$ and $p_y(x)$.

The KL-divergence between two probability mass functions $p$ and $q$ over $\mathcal{U}$ can be lower bounded by:

$$\mathrm{KL}(p||q) \geq \frac{1}{2} \left( \sum_{u \in \mathcal{U}} |p(u) - q(u)| \right)^2 \tag{15}$$

via Pinsker's inequality (Tsybakov, 2009, Section 2.8). Applying (15) to (14), we have:

$$\sum_{y \in [L]} \mathbb{P}_{\mathrm{in}}(y \mid x) \cdot \log \left( \frac{\mathbb{P}_{\mathrm{in}}(y \mid x)}{p_y(x)} \right) \geq \frac{1}{2} \left( \sum_{y \in [L]} |\mathbb{P}_{\mathrm{in}}(y \mid x) - p_y(x)| \right)^2,$$

and therefore:

$$\mathbb{E}_{(x,y)\sim\mathbb{P}_{\mathrm{in}}} \left[ \ell_{\mathrm{mc}}(y, f(x)) \right] - \mathbb{E}_{(x,y)\sim\mathbb{P}_{\mathrm{in}}} \left[ \ell_{\mathrm{mc}}(y, f^*(x)) \right] \geq \frac{1}{2} \cdot \mathbb{E}_{x\sim\mathbb{P}_{\mathrm{in}}} \left[ \left( \sum_{y \in [L]} |\mathbb{P}_{\mathrm{in}}(y \mid x) - p_y(x)| \right)^2 \right]$$

$$\geq \frac{1}{2} \left( \mathbb{E}_{x\sim\mathbb{P}_{\mathrm{in}}} \left[ \sum_{y \in [L]} |\mathbb{P}_{\mathrm{in}}(y \mid x) - p_y(x)| \right] \right)^2,$$

or

$$\mathbb{E}_{x\sim\mathbb{P}_{\mathrm{in}}} \left[ \sum_{y \in [L]} |\mathbb{P}_{\mathrm{in}}(y \mid x) - p_y(x)| \right] \leq \sqrt{2} \cdot \sqrt{\mathbb{E}_{(x,y)\sim\mathbb{P}_{\mathrm{in}}} \left[ \ell_{\mathrm{mc}}(y, f(x)) \right] - \mathbb{E}_{(x,y)\sim\mathbb{P}_{\mathrm{in}}} \left[ \ell_{\mathrm{mc}}(y, f^*(x)) \right]}.$$

**Regret Bound 2**: We expand the binary sigmoid cross-entropy loss to get:

$$\mathbb{E}_{(x,z)\sim\mathbb{P}^*} \left[ \ell_{\mathrm{bc}}(z, s(x)) \right] = \mathbb{E}_{x\sim\mathbb{P}^*} \left[ -\mathbb{P}^*(z = 1 \mid x) \cdot \log \left( p_{\perp}(x) \right) - \mathbb{P}^*(z = -1 \mid x) \cdot \log \left( 1 - p_{\perp}(x) \right) \right]$$

$$\mathbb{E}_{(x,z)\sim\mathbb{P}^*} \left[ \ell_{\mathrm{bc}}(z, s^*(x)) \right] = \mathbb{E}_{x\sim\mathbb{P}^*} \left[ -\mathbb{P}^*(z = 1 \mid x) \cdot \log \left( \mathbb{P}^*(z = 1 \mid x) \right) - \mathbb{P}^*(z = -1 \mid x) \cdot \log \left( \mathbb{P}^*(z = -1 \mid x) \right) \right],$$

and furthermore

$$\mathbb{E}_{(x,z)\sim\mathbb{P}^*} \left[ \ell_{\mathrm{bc}}(z, s(x)) \right] - \mathbb{E}_{(x,z)\sim\mathbb{P}^*} \left[ \ell_{\mathrm{bc}}(z, s^*(x)) \right]$$

$$= \mathbb{E}_{x\sim\mathbb{P}^*} \left[ \mathbb{P}^*(z = 1 \mid x) \cdot \log \left( \frac{\mathbb{P}^*(z = 1 \mid x)}{p_{\perp}(x)} \right) + \mathbb{P}^*(z = -1 \mid x) \cdot \log \left( \frac{\mathbb{P}^*(z = -1 \mid x)}{1 - p_{\perp}(x)} \right) \right]$$

$$\geq \mathbb{E}_{x\sim\mathbb{P}^*} \left[ \frac{1}{2} \left( |\mathbb{P}^*(z = 1 \mid x) - p_{\perp}(x)| + |\mathbb{P}^*(z = -1 \mid x) - (1 - p_{\perp}(x))| \right)^2 \right]$$

$$= \mathbb{E}_{x\sim\mathbb{P}^*} \left[ \frac{1}{2} \left( |\mathbb{P}^*(z = 1 \mid x) - p_{\perp}(x)| + |(1 - \mathbb{P}^*(z = 1 \mid x)) - (1 - p_{\perp}(x))| \right)^2 \right]$$

$$= 2 \cdot \mathbb{E}_{x\sim\mathbb{P}^*} \left[ |\mathbb{P}^*(z = 1 \mid x) - p_{\perp}(x)|^2 \right]$$

$$\geq 2 \cdot \left( \mathbb{E}_{x\sim\mathbb{P}^*} \left[ |\mathbb{P}^*(z = 1 \mid x) - p_{\perp}(x)| \right] \right)^2,$$

where the second step uses the bound in (15) and the last step uses Jensen's inequality. Note here that $p_{\perp}(x)$ serves as an approximation to $\mathbb{P}^*(z = 1 \mid x)$.

Taking square-root on both sides and noting that $\mathbb{P}^*(z = 1 \mid x) = \frac{\mathbb{P}_{\mathrm{in}}(x)}{\mathbb{P}_{\mathrm{in}}(x) + \mathbb{P}_{\mathrm{out}}(x)}$ completes the proof. □

## B   RELATIONSHIP BETWEEN SELECTIVE CLASSIFICATION AND LEARNING TO REJECT

There are two closely related formulations for classification problems with an abstention option: one is *selective classification* (SC), where one uses the conditional error $\mathbb{P}(y \neq h(x) \mid r(x) = 0)$

Geifman and El-Yaniv (2019)); the other is *learning to reject* (L2R), where one instead uses the joint error $\mathbb{P}(y \neq h(x), r(x) = 0)$ Ramaswamy et al. (2018).

There is a one-to-correspondence between the two formulations: owing to the constraint on $\mathbb{P}(r(x) = 1)$ in (1), it is not hard to see that:

$$\min_{h,r} \mathbb{P}(y \neq h(x) \mid r(x) = 0) \colon \mathbb{P}(r(x) = 1) \leq b$$

$$= \min_{h,r} \frac{\mathbb{P}(y \neq h(x), r(x) = 0)}{\mathbb{P}(r(x) = 0)} \colon \mathbb{P}(r(x) = 1) \leq b$$

$$= \min_{h,r,a} \frac{\mathbb{P}(y \neq h(x), r(x) = 0)}{a} \colon \mathbb{P}(r(x) = 1) \leq b, \mathbb{P}(r(x) = 0) \geq a$$

$$= \min_{h,r,a} \frac{\mathbb{P}(y \neq h(x), r(x) = 0)}{a} \colon \mathbb{P}(r(x) = 1) \leq \min(b, 1 - a)$$

Thus, *for a fixed $a$*, the problem is equivalent to (1), with a modified choice of the constraint on $\mathbb{P}(r(x) = 1)$. The Bayes-optimal classifier is thus unaffected.

## C    ALTERNATE FORMULATIONS FOR SCOD

Like us, the prior work of Xia and Bouganis (2022) also formulate SCOD as a constrained optimization problem, and consider two types of constraints: (i) a constraint on the total fraction of abstention in (3); and (ii) a constraint on the fraction of correctly classified ID samples that were accepted:

$$\min_{h,r} (1 - c_{\mathrm{fn}}) \cdot \mathbb{P}_{\mathrm{in}}(y \neq h(x), r(x) = 0) + c_{\mathrm{fn}} \cdot \mathbb{P}_{\mathrm{out}}(r(x) = 0)$$

$$\mathbb{P}_{\mathrm{in}}(r(x) = 0 \mid y = h(x)) \geq b_{\mathrm{tnr}},$$

for some budget $b_{\mathrm{tnr}}$. As acknowledged by Xia and Bouganis (2022), the second constraint can be limiting, as it only considers abstentions on the correctly classified samples (see Section 5.1 in their paper). They argue that the coverage constraint we employ is more appropriate for SCOD.

Furthermore, the Lagrangian formulation we consider in (4) with costs $c_{\mathrm{in}}$ and $c_{\mathrm{out}}$ is fairly general, and captures a wide range of SCOD formulations. For example, we can show under mild distributional assumptions that for any SCOD problem of the following form:

$$\min_{h,r} (1 - c_{\mathrm{fn}}) \cdot \mathbb{P}_{\mathrm{in}}(y \neq h(x), r(x) = 0) + c_{\mathrm{fn}} \cdot \mathbb{P}_{\mathrm{out}}(r(x) = 0)$$

$$\kappa_{00} \cdot \mathbb{P}_{\mathrm{in}}(r(x) = 0) + \kappa_{01} \cdot \mathbb{P}_{\mathrm{in}}(r(x) = 1) + \kappa_{10} \cdot \mathbb{P}_{\mathrm{out}}(r(x) = 0) + \kappa_{11} \cdot \mathbb{P}_{\mathrm{out}}(r(x) = 1) \leq b,$$

where $\kappa_{00}, \kappa_{01}, \kappa_{10}, \kappa_{11}, b \in \mathbb{R}_+$, we can formulate an equivalent objective of the form in (4), for appropriate choices of costs $c_{\mathrm{in}}$ and $c_{\mathrm{out}}$.

## D    LAGRANGIAN ANALYSIS FOR SCOD

Let $R(h, r) = \mathbb{P}_{\mathrm{in}}(y \neq h(x), r(x) = 0) + c \cdot \mathbb{P}_{\mathrm{out}}(r(x) = 0)$. We wish to solve (3), which is re-written below:

$$\min_{h,r} R(h, r) \colon \mathbb{P}_{\mathrm{te}}(r(x) = 1) \leq b.$$

The Lagrangian for this problem is given by:

$$F(h, r, \lambda) = R(h, r) + \lambda \cdot (\mathbb{P}_{\mathrm{te}}(r(x) = 1) - b),$$

where $\lambda \geq 0$ is the Lagrange multiplier. We now explicate when it is admissible to use the unconstrained Lagrangian to solve the constrained problem (3).

**Assumption D.1.** $\mathbb{P}_{\mathrm{in}}(y \mid x), \mathbb{P}_{\mathrm{in}}(x), \mathbb{P}_{\mathrm{out}}(x)$ and $\mathbb{P}_{\mathrm{te}}(x)$ are continuous in $x$.

**Theorem D.2.** *Under Assumption D.1, there exists $\lambda > 0$ such that:*

$$(h_\lambda^*, r_\lambda^*) \in \mathrm{argmin}_{h,r} F(h, r, \lambda) \implies (h_\lambda^*, r_\lambda^*) \in \mathrm{argmin}_{h,r \colon \mathbb{P}_{\mathrm{te}}(r(x)=1) \leq b} R(h, r).$$

We will find it useful to state the following lemma:

**Lemma D.3.** *Let $(h_\lambda^*, r_\lambda^*)$ be the minimizer of $F(h, r, \lambda)$ for Lagrange multiplier $\lambda \geq 0$. Then:*

$$R(h_\lambda^*, r_\lambda^*) \leq R(h, r),$$

*for all $(h, r)$ such that $\mathbb{P}_{\text{te}}(r(x) = 1) \leq \mathbb{P}_{\text{te}}(r_\lambda^*(x) = 1)$.*

*Proof.* Since $(h_\lambda^*, r_\lambda^*)$ minimizes the Lagrangian, for any $(h, r)$, $F(h_\lambda^*, r_\lambda^*, \lambda) \leq F(h, r, \lambda)$, i.e.,

$$R(h_\lambda^*, r_\lambda^*) \leq R(h, r) + \lambda \cdot (\mathbb{P}_{\text{te}}(r(x) = 1) - \mathbb{P}_{\text{te}}(r_\lambda^*(x) = 1)).$$

Since $\lambda \geq 0$, for any $(h, r)$ such that $\mathbb{P}_{\text{te}}(r(x) = 1) \leq \mathbb{P}_{\text{te}}(r_\lambda^*(x) = 1)$,

$$R(h_\lambda^*, r_\lambda^*) \leq R(h, r),$$

as desired. $\qquad\square$

We are now ready to prove Theorem D.2.

*Proof of Theorem D.2.* For a fixed $\lambda \geq 0$, the minimizer of the Lagrangian $F(h, r, \lambda)$ takes the form:

$$h_\lambda^*(x) = \text{argmax}_{y \in [L]} \mathbb{P}_{\text{in}}(y \mid x);$$

$$r_\lambda^*(x) = 1 \iff \left( \max_{y \in [L]} \mathbb{P}_{\text{in}}(y \mid x) + c - 1 \right) \cdot \frac{\mathbb{P}_{\text{in}}(x)}{\mathbb{P}_{\text{out}}(x)} < \lambda.$$

The abstention rate for $r_\lambda^*(x)$ can then be written as:

$$\mathbb{P}_{\text{te}}(r_\lambda^*(x) = 1) = \int_{H(x) < \lambda} \mathbb{P}_{\text{te}}(x) dx.$$

where $H(x) = \left( \max_{y \in [L]} \mathbb{P}_{\text{in}}(y \mid x) + c - 1 \right) \cdot \frac{\mathbb{P}_{\text{in}}(x)}{\mathbb{P}_{\text{out}}(x)}$.

Since $\mathbb{P}_{\text{in}}(y \mid x), \mathbb{P}_{\text{in}}(x), \mathbb{P}_{\text{out}}(x)$ are continuous in $x$, $H(x)$ is continuous in $x$. Furthermore, since the density $\mathbb{P}_{\text{te}}(x)$ is also continuous, we can always find a $\lambda \geq 0$ for which $\mathbb{P}_{\text{te}}(r_\lambda^*(x) = 1) = b$. Applying Lemma D.3 with this choice of $\lambda$, we then have that $R(h_\lambda^*, r_\lambda^*) \leq R(h, r)$ for all $(h, r)$ such that $\mathbb{P}_{\text{te}}(r(x) = 1) \leq b$. $\qquad\square$

When the underlying distributions are discrete or mixed, there may be budgets $b$ for which no equivalent Lagrange multiplier $\lambda$ exists. In such cases, one may choose the multiplier with coverage closest to $b$ and solve a relaxation to (3).

# E    GENERALIZATION ANALYSIS

For labeled set $S_{\text{in}} \sim \mathbb{P}_{\text{in}}$, and unlabeled set $S_{\text{out}} \sim \mathbb{P}_{\text{out}}$, we denote:

$$S^* = \{(x, 1) : (x, y) \in S_{\text{in}}\} \cup \{(x, -1) : x \in S_{\text{out}}\}.$$

Let $n_{\text{in}} = |S_{\text{in}}|$, $n_{\text{out}} = |S_{\text{out}}|$ and $n^* = n_{\text{in}} + n_{\text{out}}$. We denote the expected risks by:

$$R_{\text{mc}}(f) = \mathbb{E}_{(x,y) \sim \mathbb{P}_{\text{in}}} [\ell_{\text{mc}}(y, f(x))]; \quad R_{\text{bc}}(s) = \mathbb{E}_{(x,z) \sim \mathbb{P}^*} [\ell_{\text{bc}}(z, s(x))],$$

and their empirical counter-parts by:

$$\hat{R}_{\text{mc}}(f) = \frac{1}{n_{\text{in}}} \sum_{(x,y) \in S_{\text{in}}} \ell_{\text{mc}}(y, f(x)); \quad \hat{R}_{\text{bc}}(s) = \frac{1}{n^*} \sum_{(x,z) \in S^*} \ell_{\text{bc}}(z, s(x)).$$

Let $\mathcal{F}$ be a hypothesis class of bounded scorers of the form $f : \mathcal{X} \to \mathbb{R}^L$ and $\mathcal{G}$ be a class of bounded scorers $s : \mathcal{X} \to \mathbb{R}$. Let $\mathcal{N}(\mathcal{F}, \epsilon)$ denote the covering number for $\mathcal{F}$ with the $\infty$-norm, and $\mathcal{N}(\mathcal{G}, \epsilon)$ similarly denote the covering number for $\mathcal{G}$. Let $(\tilde{f}, \tilde{s})$ be the minimizer of the decoupled loss $R_{\text{mc}}(f) + R_{\text{bc}}(s)$ over $\mathcal{F} \times \mathcal{G}$, and $(f^*, s^*)$ be the minimizers over all measurable functions.

**Lemma E.1.** *Suppose $\ell_{\mathrm{mc}}, \ell_{\mathrm{bc}}$ correspond to the softmax and sigmoid cross-entropy losses, with $\ell_{\mathrm{mc}}(\cdot, \cdot) \leq B_{\mathrm{mc}}$ and $\ell_{\mathrm{bc}}(\cdot, \cdot) \leq B_{\mathrm{bc}}$. Let $(\hat{f}, \hat{s})$ be the minimizer of the empirical decoupled loss in (10), i.e., of $\hat{R}_{\mathrm{mc}}(f) + \hat{R}_{\mathrm{bc}}(s)$ over $\mathcal{F} \times \mathcal{G}$. Let $\hat{p}_y(x) = \frac{\exp(\hat{f}_y(x))}{\sum_{y'} \exp(\hat{f}_{y'}(x))}$ and $\hat{p}_{\perp}(x) = \frac{1}{1 + \exp(-\hat{s}(x))}$, with probability at least $1 - \delta$ over draw of $S_{\mathrm{in}}$ and $S_{\mathrm{out}}$:*

$$\mathbb{E}_{x \sim \mathbb{P}_{\mathrm{in}}} \left[ \sum_{y \in [L]} |\hat{p}_y(x) - \mathbb{P}_{\mathrm{in}}(y \mid x)| \right] \leq 2 \left( 2 \cdot \inf_{\epsilon > 0} \left\{ \epsilon + B_{\mathrm{mc}} \sqrt{\frac{2 \cdot \log \mathcal{N}(\mathcal{G}, \epsilon/L)}{n_{\mathrm{in}}}} \right\} + \mathcal{O} \left( \sqrt{\frac{\log(1/\delta)}{n_{\mathrm{in}}}} \right) \right)^{1/2}$$
$$+ \sqrt{2} \cdot \sqrt{R_{\mathrm{mc}}(\tilde{f}) - R_{\mathrm{mc}}(f^*)};$$

$$\mathbb{E}_{x \sim \mathbb{P}^*} \left[ \left| \hat{p}_{\perp}(x) - \frac{\mathbb{P}_{\mathrm{in}}(x)}{\mathbb{P}_{\mathrm{in}}(x) + \mathbb{P}_{\mathrm{out}}(x)} \right| \right] \leq 2 \left( 2 \cdot \inf_{\epsilon > 0} \left\{ \epsilon + B_{\mathrm{bc}} \sqrt{\frac{2 \cdot \log \mathcal{N}(\mathcal{G}, \epsilon)}{n^*}} \right\} + \mathcal{O} \left( \sqrt{\frac{\log(1/\delta)}{n^*}} \right) \right)^{1/2}$$
$$+ \sqrt{2} \cdot \sqrt{R_{\mathrm{bc}}(\tilde{s}) - R_{\mathrm{bc}}(s^*)}.$$

The proof uses the following generalization bounds based on uniform convergence (Shalev-Shwartz and Ben-David, 2014).

**Lemma E.2.** *Suppose the losses $\ell_{\mathrm{mc}}(\cdot, \cdot) \leq B_{\mathrm{mc}}$ and $\ell_{\mathrm{bc}}(\cdot, \cdot) \leq B_{\mathrm{bc}}$. For any $\delta \in (0, 1)$, with probability at least $1 - \delta$ over draw of $S_{\mathrm{in}}$ and $S_{\mathrm{out}}$, for any $f' \in \mathcal{F}$ and $s' \in \mathcal{G}$:*

$$\left| R_{\mathrm{mc}}(f') - \hat{R}_{\mathrm{mc}}(f') \right| \leq 2 \cdot \inf_{\epsilon > 0} \left\{ \epsilon + B_{\mathrm{mc}} \sqrt{\frac{2 \cdot \log (\mathcal{N}(\mathcal{F}, \epsilon/L))}{n_{\mathrm{in}}}} \right\} + \mathcal{O} \left( \sqrt{\frac{\log(1/\delta)}{n_{\mathrm{in}}}} \right);$$

$$\left| R_{\mathrm{bc}}(s') - \hat{R}_{\mathrm{bc}}(s') \right| \leq 2 \cdot \inf_{\epsilon > 0} \left\{ \epsilon + B_{\mathrm{bc}} \sqrt{\frac{2 \cdot \log (\mathcal{N}(\mathcal{G}, \epsilon))}{n^*}} \right\} + \mathcal{O} \left( \sqrt{\frac{\log(1/\delta)}{n^*}} \right).$$

*Proof of Lemma E.2.* We prove the second bound. The first bound follows through similar arguments. Let $\mathcal{L} = \{(x, y) \mapsto \ell_{\mathrm{bc}}(y, s(x)) \mid s \in \mathcal{G}\}$ be the class of sigmoid cross-entropy losses induced by hypothesis class $\mathcal{G}$. Let $\hat{\mathcal{R}}(\mathcal{L})$ denote the empirical Radamacher complexity of $\mathcal{L}$. Then a standard two-sided Radamacher complexity bound gives us that with probability at least $1 - \delta$ (see C. Scott, UMich EECS 598: Statistical Learning Theory, Winter 2014, Topic 10, Theorem 2):

$$\sup_{s' \in \mathcal{G}} \left| R_{\mathrm{bc}}(s') - \hat{R}_{\mathrm{bc}}(s') \right| \leq 2 \cdot \hat{\mathcal{R}}(\mathcal{L}) + \mathcal{O} \left( \sqrt{\frac{\log(1/\delta)}{n^*}} \right).$$

We next bound $\hat{\mathcal{R}}(\mathcal{L})$ in terms of the covering number of $\mathcal{L}$. Fix $\epsilon > 0$. Let $\tilde{\mathcal{G}}$ be an $\epsilon$-cover for $\mathcal{G}$ under the $\infty$-norm. By our assumption there exists such a cover with size $|\tilde{\mathcal{G}}| = \mathcal{N}(\mathcal{G}, \epsilon)$. This implies that for any $s \in \mathcal{G}$, there exists a $\tilde{s} \in \tilde{\mathcal{G}}$ such that $\sup_x |s(x) - \tilde{s}(x)| \leq \epsilon$. Since the loss $\ell_{\mathrm{bc}}$ is 1-Lipschitz in its second argument, this further implies that for any $s \in \mathcal{G}$, there exists a $\tilde{s} \in \tilde{\mathcal{G}}$ such that $\sup_{(x,y)} |\ell_{\mathrm{bc}}(y, s(x)) - \ell_{\mathrm{bc}}(y, \tilde{s}(x))| \leq \epsilon$.

We then have:

$$\hat{\mathcal{R}}(\mathcal{G}) = \mathbb{E}_{\sigma} \left[ \sup_{s \in \mathcal{G}} \frac{1}{n^*} \sum_{i=1}^{n^*} \sigma_i \cdot \ell_{\mathrm{bc}}(y_i, s(x_i)) \right]$$

$$= \mathbb{E}_{\sigma} \left[ \sup_{s \in \mathcal{G}} \frac{1}{n^*} \sum_{i=1}^{n^*} \sigma_i \cdot \ell_{\mathrm{bc}}(y_i, \tilde{s}(x_i)) + \frac{1}{n^*} \sum_{i=1}^{n^*} \sigma_i \cdot (\ell_{\mathrm{bc}}(y_i, s(x_i)) - \ell_{\mathrm{bc}}(y_i, \tilde{s}(x_i))) \right]$$

$$\leq \mathbb{E}_{\sigma} \left[ \sup_{s \in \mathcal{G}} \frac{1}{n^*} \sum_{i=1}^{n^*} \sigma_i \cdot \ell_{\mathrm{bc}}(y_i, \tilde{s}(x_i)) + \frac{1}{n^*} \left( \sum_{i=1}^{n^*} |\sigma_i| \right) \cdot |\ell_{\mathrm{bc}}(y_i, s(x_i)) - \ell_{\mathrm{bc}}(y_i, \tilde{s}(x_i))| \right]$$

$$\leq \mathbb{E}_{\sigma} \left[ \sup_{s \in \mathcal{G}} \frac{1}{n^*} \sum_{i=1}^{n^*} \sigma_i \cdot \ell_{\mathrm{bc}}(y_i, \tilde{s}(x_i)) + \epsilon \right],$$

where $\sigma$ is a random variable drawn uniformly from $\{-1, +1\}^{n^*}$. By Massart's lemma (Shalev-Shwartz & Ben-David, 2014, Lemma 26.8) and using the fact that $\ell_{\mathrm{bc}}(\cdot, \cdot) \leq B_{\mathrm{bc}}$, we have:

$$\hat{\mathcal{R}}(\mathcal{G}) \leq \max_{s \in \tilde{\mathcal{G}}} \sqrt{\sum_{i=1}^{n^*} (\ell_{\mathrm{bc}}(y_i, \tilde{s}(x_i))^2 \cdot \frac{\sqrt{2 \cdot \log\left(|\tilde{\mathcal{G}}|\right)}}{n^*}} + \epsilon$$

$$\leq B_{\mathrm{bc}} \cdot \sqrt{n^*} \cdot \frac{\sqrt{2 \cdot \log \mathcal{N}(\mathcal{G}, \epsilon)}}{n^*} + \epsilon$$

$$= B_{\mathrm{bc}} \cdot \sqrt{\frac{2 \cdot \log \mathcal{N}(\mathcal{G}, \epsilon)}{n^*}} + \epsilon.$$

This holds for any $\epsilon > 0$. Taking an infimum over $\epsilon$ completes the proof. $\qquad \square$

*Proof of Lemma E.1.* Applying Lemma 4.2 to $\hat{f}$ and $\hat{s}$, we have:

$$\mathbb{E}_{x \sim \mathbb{P}_{\mathrm{in}}} \Big[ \sum_{y \in [L]} \big| \hat{p}_y(x) - \mathbb{P}_{\mathrm{in}}(y \mid x) \big| \Big]$$

$$\leq \sqrt{2} \cdot \sqrt{R_{\mathrm{mc}}(\hat{f}) - R_{\mathrm{mc}}(f^*)}$$

$$\leq \sqrt{2} \cdot \sqrt{R_{\mathrm{mc}}(\hat{f}) - R_{\mathrm{mc}}(\tilde{f})} + \sqrt{2} \cdot \sqrt{R_{\mathrm{mc}}(\tilde{f}) - R_{\mathrm{mc}}(f^*)}$$

$$= \sqrt{2} \cdot \sqrt{R_{\mathrm{mc}}(\hat{f}) - \hat{R}_{\mathrm{mc}}(\tilde{f}) + \hat{R}_{\mathrm{mc}}(\tilde{f}) - R_{\mathrm{mc}}(\tilde{f})} + \sqrt{2} \cdot \sqrt{R_{\mathrm{mc}}(\tilde{f}) - R_{\mathrm{mc}}(f^*)}$$

$$\leq \sqrt{2} \cdot \sqrt{R_{\mathrm{mc}}(\hat{f}) - \hat{R}_{\mathrm{mc}}(\hat{f}) + \hat{R}_{\mathrm{mc}}(\tilde{f}) - R_{\mathrm{mc}}(\tilde{f})} + \sqrt{2} \cdot \sqrt{R_{\mathrm{mc}}(\tilde{f}) - R_{\mathrm{mc}}(f^*)}$$

$$\leq \sqrt{2} \cdot \sqrt{|R_{\mathrm{mc}}(\hat{f}) - \hat{R}_{\mathrm{mc}}(\hat{f})| + |R_{\mathrm{mc}}(\tilde{f}) - \hat{R}_{\mathrm{mc}}(\tilde{f})|} + \sqrt{2} \cdot \sqrt{R_{\mathrm{mc}}(\tilde{f}) - R_{\mathrm{mc}}(f^*)}$$

$$\leq \sup_{f' \in \mathcal{F}} \sqrt{2} \cdot \sqrt{2 \cdot |R_{\mathrm{mc}}(f') - \hat{R}_{\mathrm{mc}}(f')|} + \sqrt{2} \cdot \sqrt{R_{\mathrm{mc}}(\tilde{f}) - R_{\mathrm{mc}}(f^*)},$$

where the third step uses the fact that $\hat{R}_{\mathrm{mc}}(\hat{f}) \leq \hat{R}_{\mathrm{mc}}(\tilde{f})$. This follows from the fact that $(\hat{f}, \hat{s})$ is a minimizer of the empirical decoupled loss in (10), i.e., of $\hat{R}_{\mathrm{mc}}(f) + \hat{R}_{\mathrm{bc}}(s)$, over $\mathcal{F} \times \mathcal{G}$; since the losses are decoupled, $\hat{f}$ is a minimizer of $\hat{R}_{\mathrm{mc}}(f)$ over $\mathcal{F}$. We similarly have:

$$\mathbb{E}_{x \sim \mathbb{P}^*} \left[ \left| \hat{p}_\perp(x) - \frac{\mathbb{P}_{\mathrm{in}}(x)}{\mathbb{P}_{\mathrm{in}}(x) + \mathbb{P}_{\mathrm{out}}(x)} \right| \right]$$

$$\leq \sup_{s' \in \mathcal{G}} \sqrt{2} \cdot \sqrt{2 \cdot |R_{\mathrm{bc}}(s') - \hat{R}_{\mathrm{bc}}(s')|} + \sqrt{2} \cdot \sqrt{R_{\mathrm{mc}}(\tilde{s}) - R_{\mathrm{mc}}(s^*)}.$$

Substituting the right-hand sides with the bound from Lemma E.2 completes the proof. $\qquad \square$

## F  Coupled Loss for SCOD

Our second loss function seeks to learn an augmented scorer $\bar{f} \colon \mathcal{X} \to \mathbb{R}^{L+1}$, with the additional score corresponding to a "reject class", denoted by $\perp$, and is based on the following simple observation: define

$$z_{y'}(x) = \begin{cases} (1 - c_{\mathrm{in}} - c_{\mathrm{out}}) \cdot \mathbb{P}_{\mathrm{in}}(y \mid x) & \text{if } y' \in [L] \\ (1 - 2 \cdot c_{\mathrm{in}} - c_{\mathrm{out}}) + c_{\mathrm{out}} \cdot \frac{\mathbb{P}_{\mathrm{out}}(x)}{\mathbb{P}_{\mathrm{in}}(x)} & \text{if } y' = \perp, \end{cases}$$

and let $\zeta_{y'}(x) = \frac{z_{y'}(x)}{Z(x)}$ for $Z(x) \doteq \sum_{y'' \in [L] \cup \{\perp\}} z_{y''}(x)$. Now suppose that one has an estimate $\hat{\zeta}$ of $\zeta$. This yields an alternate plug-in estimator of the Bayes-optimal SCOD rule (5):

$$\hat{r}(x) = 1 \iff \max_{y' \in [L]} \hat{\zeta}_{y'}(x) < \hat{\zeta}_\perp(x). \tag{16}$$

One may readily estimate $\zeta_{y'}$ with a standard multi-class loss $\ell_{\mathrm{mc}}$, with suitable modification:

$$\mathbb{E}_{(x,y) \sim \mathbb{P}_{\mathrm{in}}} \big[ \ell_{\mathrm{mc}}(y, \bar{f}(x)) \big] + (1 - c_{\mathrm{in}}) \cdot \mathbb{E}_{x \sim \mathbb{P}_{\mathrm{in}}} \big[ \ell_{\mathrm{mc}}(\perp, \bar{f}(x)) \big] + c_{\mathrm{out}} \cdot \mathbb{E}_{x \sim \mathbb{P}_{\mathrm{out}}} \big[ \ell_{\mathrm{mc}}(\perp, \bar{f}(x)) \big]. \tag{17}$$

Compared to the decoupled loss (10), the key difference is that the penalties on the rejection logit $\bar{f}_\perp(x)$ involve the classification logits as well.

### F.1 RELATION TO EXISTING LOSSES

Equation 10 generalises several existing proposals in the SC and OOD detection literature. In particular, it reduces to the loss proposed in Verma and Nalisnick (2022), when $\mathbb{P}_{\text{in}} = \mathbb{P}_{\text{out}}$, i.e., when one only wishes to abstain on low confidence ID samples. Interestingly, this also corresponds to the decoupled loss for OOD detection in Bitterwolf et al. (2022); crucially, however, they reject only based on whether $\bar{f}_{\perp}(x) < 0$, rather than comparing $\bar{f}_{\perp}(x)$ and $\max_{y' \in [L]} \bar{f}_{y'}(x)$. The latter is essential to match the Bayes-optimal predictor in (5). Similarly, the coupled loss in (17) reduces to the *cost-sensitive softmax cross-entropy* in Mozannar and Sontag (2020) when $c_{\text{out}} = 0$, and the OOD detection loss of Thulasidasan et al. (2021) when $c_{\text{in}} = 0, c_{\text{out}} = 1$.

## G   TECHNICAL DETAILS: ESTIMATING THE OOD MIXING WEIGHT $\pi_{\text{mix}}$

To obtain the latter, we apply a simple transformation as follows.

**Lemma G.1.** *Suppose* $\mathbb{P}_{\text{mix}} = \pi_{\text{mix}} \cdot \mathbb{P}_{\text{in}} + (1 - \pi_{\text{mix}}) \cdot \mathbb{P}_{\text{out}}$ *with* $\pi_{\text{mix}} < 1$. *Then, if* $\mathbb{P}_{\text{in}}(x) > 0$,

$$\frac{\mathbb{P}_{\text{out}}(x)}{\mathbb{P}_{\text{in}}(x)} = \frac{1}{1 - \pi_{\text{mix}}} \cdot \left( \frac{\mathbb{P}_{\text{mix}}(x)}{\mathbb{P}_{\text{in}}(x)} - \pi_{\text{mix}} \right).$$

The above transformation requires knowing the mixing proportion $\pi_{\text{mix}}$ of inlier samples in the unlabeled dataset. However, as it measures the fraction of OOD samples during deployment, $\pi_{\text{mix}}$ is typically *unknown*. We may however estimate this with **(A2)**. Observe that for a strictly inlier example $x \in S_{\text{in}}^*$, we have $\frac{\mathbb{P}_{\text{mix}}(x)}{\mathbb{P}_{\text{in}}(x)} = \pi_{\text{mix}}$, i.e., $\exp(-\hat{s}(x)) \approx \pi_{\text{mix}}$. Therefore, we can estimate

$$\hat{s}_{\text{ood}}(x) = \left( \frac{1}{1 - \hat{\pi}_{\text{mix}}} \cdot (\exp(-\hat{s}(x)) - \hat{\pi}_{\text{mix}}) \right)^{-1} \quad \text{where} \quad \hat{\pi}_{\text{mix}} = \frac{1}{|S_{\text{in}}^*|} \sum_{x \in S_{\text{in}}^*} \exp(-\hat{s}(x)).$$

We remark here that this problem is roughly akin to class prior estimation for PU learning (Garg et al., 2021), and noise rate estimation for label noise (Patrini et al., 2017). As in those literatures, estimating $\pi_{\text{mix}}$ without any assumptions is challenging. Our assumption on the existence of a Strict Inlier set $S_{\text{in}}^*$ is analogous to assuming the existence of a golden label set in the label noise literature (Hendrycks et al., 2018).

*Proof of Lemma G.1.* Expanding the right-hand side, we have:

$$\frac{1}{1 - \pi_{\text{mix}}} \cdot \left( \frac{\mathbb{P}_{\text{mix}}(x)}{\mathbb{P}_{\text{in}}(x)} - \pi_{\text{mix}} \right) = \frac{1}{1 - \pi_{\text{mix}}} \cdot \left( \frac{\pi_{\text{mix}} \cdot \mathbb{P}_{\text{in}}(x) + (1 - \pi_{\text{mix}}) \cdot \mathbb{P}_{\text{out}}(x)}{\mathbb{P}_{\text{in}}(x)} - \pi_{\text{mix}} \right)$$
$$= \frac{\mathbb{P}_{\text{out}}(x)}{\mathbb{P}_{\text{in}}(x)},$$

as desired. □

## H   TECHNICAL DETAILS: PLUG-IN ESTIMATORS WITH AN ABSTENTION BUDGET

Recall that the constrained SCOD objective stated in (3) is

$$\min_{h,r} (1 - c_{\text{fn}}) \cdot \mathbb{P}_{\text{in}}(y \neq h(x), r(x) = 0) + c_{\text{fn}} \cdot \mathbb{P}_{\text{out}}(r(x) = 0) : \mathbb{P}_{\text{te}}(r(x) = 1) \leq b_{\text{rej}}. \quad (18)$$

The corresponding Lagrangian is

$$\min_{h,r} \max_{\lambda} F(h, r; \lambda)$$

where

$$F(h, r; \lambda) \doteq (1 - c_{\text{fn}}) \cdot \mathbb{P}_{\text{in}}(y \neq h(x), r(x) = 0) + c_{\text{fn}} \cdot \mathbb{P}_{\text{out}}(r(x) = 0) + \lambda \cdot \mathbb{P}_{\text{te}}(r(x) = 1) - \lambda \cdot b_{\text{rej}}$$
$$= (1 - c_{\text{fn}}) \cdot \mathbb{P}_{\text{in}}(y \neq h(x), r(x) = 0) + c_{\text{fn}} \cdot \mathbb{P}_{\text{out}}(r(x) = 0) + \lambda \cdot \pi_{\text{in}}^* \cdot \mathbb{P}_{\text{in}}(r(x) = 1) +$$
$$= \lambda \cdot (1 - \pi_{\text{in}}^*) \cdot \mathbb{P}_{\text{out}}(r(x) = 1) - \lambda \cdot b_{\text{rej}}$$

$$
\begin{aligned}
&= (1 - c_{\mathrm{fn}}) \cdot \mathbb{P}_{\mathrm{in}}(y \neq h(x), r(x) = 0) + c_{\mathrm{fn}} \cdot \mathbb{P}_{\mathrm{out}}(r(x) = 0) + \lambda \cdot \pi_{\mathrm{in}}^* \cdot \mathbb{P}_{\mathrm{in}}(r(x) = 1) + \\
&= \lambda \cdot (1 - \pi_{\mathrm{in}}^*) - \lambda \cdot (1 - \pi_{\mathrm{in}}^*) \cdot \mathbb{P}_{\mathrm{out}}(r(x) = 0) - \lambda \cdot b_{\mathrm{rej}} \\
&= (1 - c_{\mathrm{fn}}) \cdot \mathbb{P}_{\mathrm{in}}(y \neq h(x), r(x) = 0) + c_{\mathrm{in}}(\lambda) \cdot \mathbb{P}_{\mathrm{in}}(r(x) = 1) + \\
&= c_{\mathrm{out}}(\lambda) \cdot \mathbb{P}_{\mathrm{out}}(r(x) = 0) + \nu_\lambda,
\end{aligned}
\tag{19}
$$

where in the last line we define

$$
\begin{aligned}
c_{\mathrm{in}}(\lambda) &= \lambda \cdot \pi_{\mathrm{in}}^* \\
c_{\mathrm{out}}(\lambda) &= c_{\mathrm{fn}} - \lambda \cdot (1 - \pi_{\mathrm{in}}^*) \\
\nu_\lambda &= \lambda \cdot (1 - \pi_{\mathrm{in}}^*) - \lambda \cdot b_{\mathrm{rej}}.
\end{aligned}
$$

Solving (19) requires optimising over both $(h, r)$ and $\lambda$. Suppose momentarily that $\lambda$ is fixed. Then, $F(h, r; \lambda)$ is exactly a scaled version of the soft-penalty objective (4). Thus, we can use Algorithm 1 to construct a plug-in classifier that minimizes the above joint risk. To find the optimal $\lambda$, we only need to implement the surrogate minimisation step in Algorithm 1 *once* to estimate the relevant probabilities. We can then construct multiple plug-in classifiers for different values of $\lambda$, and perform an inexpensive threshold search: amongst the classifiers satisfying the budget constraint, we pick the one that minimises (19).

The above requires estimating $\pi_{\mathrm{in}}^*$, the fraction of inliers observed during deployment. Following **(A2)**, one plausible estimate is $\pi_{\mathrm{mix}}$, the fraction of inliers in the "wild" mixture set $S_{\mathrm{mix}}$. In some industry production settings, it may be reasonable to estimate $\pi_{\mathrm{in}}^*$, through, for example, inspection of logged data. This is the setting we assume in our experiments.

**Remark.** The previous work of Katz-Samuels et al. (2022) for OOD detection also seeks to solve an optimization problem with explicit constraints on abstention rates. However, there are some subtle, but important, technical differences between their formulation and ours.

Like us, Katz-Samuels et al. (2022) also seek to jointly learn a classifier and an OOD scorer, with constraints on the classification and abstention rates, given access to samples from $\mathbb{P}_{\mathrm{in}}$ and $\mathbb{P}_{\mathrm{mix}}$. For a joint classifier $h : \mathcal{X} \to [L]$ and rejector $r : \mathcal{X} \to \{0, 1\}$, their formulation can be written as:

$$
\min_h \ \mathbb{P}_{\mathrm{out}}(r(x) = 0) \tag{20}
$$
$$
\text{s.t.} \quad \mathbb{P}_{\mathrm{in}}(r(x) = 1) \leq \kappa
$$
$$
\mathbb{P}_{\mathrm{in}}(h(x) \neq y, \, r(x) = 0) \leq \tau,
$$

for given targets $\kappa, \tau \in (0, 1)$.

While $\mathbb{P}_{\mathrm{out}}$ is not directly available, Katz-Samuels et al. provide a simple solution to solving (20) using only access to $\mathbb{P}_{\mathrm{mix}}$ and $\mathbb{P}_{\mathrm{in}}$. They show that under some mild assumptions, replacing $\mathbb{P}_{\mathrm{out}}$ with $\mathbb{P}_{\mathrm{mix}}$ in the above problem does not alter the optimal solution. The intuition behind this is that when the first constraint on the inlier abstention rate is satisfied with equality, we have $\mathbb{P}_{\mathrm{mix}}(r(x) = 0) = \pi_{\mathrm{mix}} \cdot (1 - c_{\mathrm{in}}) + (1 - \pi_{\mathrm{mix}}) \cdot \mathbb{P}_{\mathrm{out}}(r(x) = 0)$, and minimizing this objective is equivalent to minimizing the OOD objective in (20).

This simple trick of replacing $\mathbb{P}_{\mathrm{out}}$ with $\mathbb{P}_{\mathrm{mix}}$ will only work when we have an explicit constraint on the inlier abstention rate, and will not work for the formulation we are interested in (19). This is because in our formulation, we impose a budget on the overall abstention rate (as this is a more intuitive quantity that a practitioner may want to constraint), and do not explicitly control the abstention rate on $\mathbb{P}_{\mathrm{in}}$.

In comparison to Katz-Samuels et al. (2022), the plug-in based approach we prescribe is more general, and can be applied to optimize any objective that involves as a weighted combination of the mis-classification error and the abstention rates on the inlier and OOD samples. This includes both the budget-constrained problem we consider in (19), and the constrained problem of Katz-Samuels et al. in (20).

## I   ILLUSTRATING THE FAILURE OF MSP FOR OOD DETECTION

### I.1   MSP FAILS FOR OPEN-SET RECOGNITION

We show that MSP may result in *arbitrarily bad* rejection decisions even for the special case of OOD detection wherein there is a strong relationship between $\mathbb{P}_{\mathrm{in}}$ and $\mathbb{P}_{\mathrm{out}}$ that *a-priori* would

appear favourable to the MSP. Specifically, given some distribution $\mathbb{P}_{te}$ over $\mathcal{X} \times \mathcal{Y}$, consider the *open-set classification* (*OSC*) setting (Scheirer et al., 2013; Vaze et al., 2021): during training, one only observes samples from a distribution $\mathbb{P}_{in}$ over $\mathcal{X} \times \mathcal{Y}_{in}$, where $\mathcal{Y}_{in} \subset \mathcal{Y}$. Here, $\mathbb{P}_{in}$ is a restriction of $\mathbb{P}_{te}$ to a subset of labels. At evaluation time, one seeks to accurately classify samples possessing these labels, while rejecting samples with unobserved labels $\mathcal{Y} - \mathcal{Y}_{in}$.

Under this setup, thresholding $\max_{y \in \mathcal{Y}_{in}} \mathbb{P}_{in}(y \mid x)$ might appear a reasonable approach. However, we now demonstrate that it may lead to arbitrarily poor decisions. In what follows, for simplicity we consider the OSC problem wherein $\mathcal{Y}_{in} = \mathcal{Y} - \{L\}$, so that there is only one label unobserved in the in-distribution sample. Further, we focus on the setting where $c_{in} + c_{out} = 1$. We have the following.

**Lemma I.1.** *Under the open-set setting, the Bayes-optimal classifier for the SCOD problem is:*

$$r^*(x) = 1 \iff \mathbb{P}_{te}(L \mid x) > t^*_{osc} \iff \max_{y' \neq L} \mathbb{P}_{in}(y' \mid x) \geq \frac{1}{1 - t^*_{osc}} \cdot \max_{y' \neq L} \mathbb{P}_{te}(y' \mid x),$$

*where $t^*_{osc} \doteq F\left( \frac{c_{in} \cdot \mathbb{P}_{te}(y=L)}{c_{out} \cdot \mathbb{P}_{te}(y \neq L)} \right)$ for $F \colon z \mapsto z/(1+z)$.*

Lemma I.1 shows that the optimal decision is to reject when the maximum softmax probability (with respect to $\mathbb{P}_{in}$) is *higher* than some (sample-dependent) threshold. This is the precise *opposite* of the MSP baseline, which rejects when the maximum probability is *lower* than some threshold. What is the reason for this stark discrepancy? Intuitively, the issue is that we would like to threshold $\mathbb{P}_{te}(y \mid x)$, *not* $\mathbb{P}_{in}(y \mid x)$; however, these two distributions may not align, as the latter includes a normalisation term that causes unexpected behaviour when we threshold. We make this concrete with a simple example; see also Figure 1 for an illustration.

*Example* I.2 (*Failure of MSP baseline*). Consider a setting where the class probabilities $\mathbb{P}_{te}(y' \mid x)$ are equal for all the known classes $y' \neq L$. This implies that $\mathbb{P}_{in}(y' \mid x) = \frac{1}{L-1}, \forall y' \neq L$. The Bayes-optimal classifier rejects a sample when $\mathbb{P}_{te}(L \mid x) > \frac{c_{in}}{c_{in}+c_{out}}$. On the other hand, MSP rejects a sample iff the threshold $t_{msp} < \frac{1}{L-1}$. Notice that the rejection decision is *independent* of the unknown class density $\mathbb{P}_{te}(L \mid x)$, and therefore will not agree with the Bayes-optimal classifier in general. The following lemma formalizes this observation.

**Lemma I.3.** *Pick any $t_{msp} \in (0, 1)$, and consider the corresponding MSP baseline which rejects $x \in \mathcal{X}$ iff $\max_{y \neq L} \mathbb{P}_{in}(y \mid x) < t_{msp}$. Then, there exists a class-probability function $\mathbb{P}_{te}(y \mid x)$ for which the Bayes-optimal rejector $\mathbb{P}_{te}(L \mid x) > t^*_{osc}$ disagrees with MSP $\forall t_{msp} \in (0, 1)$.*

*Proof of Lemma I.1.* Recall that in open-set classification, the outlier distribution is $\mathbb{P}_{out}(x) = \mathbb{P}_{te}(x \mid y = L)$, while the training distribution is

$$\mathbb{P}_{in}(x \mid y) = \mathbb{P}_{te}(x \mid y)$$
$$\pi_{in}(y) = \mathbb{P}_{in}(y)$$
$$= \frac{1(y \neq L)}{1 - \pi_{te}(L)} \cdot \pi_{te}(y).$$

We will find it useful to derive the following quantities.

$$\mathbb{P}_{in}(x, y) = \pi_{in}(y) \cdot \mathbb{P}_{in}(x \mid y)$$
$$= \frac{1(y \neq L)}{1 - \pi_{te}(L)} \cdot \pi_{te}(y) \cdot \mathbb{P}_{te}(x \mid y)$$
$$= \frac{1(y \neq L)}{1 - \pi_{te}(L)} \cdot \mathbb{P}_{te}(x, y)$$
$$\mathbb{P}_{in}(x) = \sum_{y \in [L]} \mathbb{P}_{in}(x, y)$$
$$= \sum_{y \in [L]} \pi_{in}(y) \cdot \mathbb{P}_{in}(x \mid y)$$
$$= \frac{1}{1 - \pi_{te}(L)} \sum_{y \neq L} \pi_{te}(y) \cdot \mathbb{P}_{te}(x \mid y)$$

$$= \frac{1}{1 - \pi_{\text{te}}(L)} \sum_{y \neq L} \mathbb{P}_{\text{te}}(y \mid x) \cdot \mathbb{P}_{\text{te}}(x)$$

$$= \frac{\mathbb{P}_{\text{te}}(y \neq L \mid x)}{1 - \pi_{\text{te}}(L)} \cdot \mathbb{P}_{\text{te}}(x)$$

$$\mathbb{P}_{\text{in}}(y \mid x) = \frac{\mathbb{P}_{\text{in}}(x, y)}{\mathbb{P}_{\text{in}}(x)}$$

$$= \frac{1(y \neq L)}{1 - \pi_{\text{te}}(L)} \cdot \frac{1 - \pi_{\text{te}}(L)}{\mathbb{P}_{\text{te}}(y \neq L \mid x)} \cdot \frac{\mathbb{P}_{\text{te}}(x, y)}{\mathbb{P}_{\text{te}}(x)}$$

$$= \frac{1(y \neq L)}{\mathbb{P}_{\text{te}}(y \neq L \mid x)} \cdot \mathbb{P}_{\text{te}}(y \mid x).$$

The first part follows from standard results in cost-sensitive learning (Elkan, 2001):

$$
\begin{aligned}
r^*(x) = 1 &\iff c_{\text{in}} \cdot \mathbb{P}_{\text{in}}(x) - c_{\text{out}} \cdot \mathbb{P}_{\text{out}}(x) < 0 \\
&\iff c_{\text{in}} \cdot \mathbb{P}_{\text{in}}(x) < c_{\text{out}} \cdot \mathbb{P}_{\text{out}}(x) \\
&\iff c_{\text{in}} \cdot \mathbb{P}_{\text{te}}(x \mid y \neq L) < c_{\text{out}} \cdot \mathbb{P}_{\text{te}}(x \mid y = L) \\
&\iff c_{\text{in}} \cdot \mathbb{P}_{\text{te}}(y \neq L \mid x) \cdot \mathbb{P}_{\text{te}}(y = L) < c_{\text{out}} \cdot \mathbb{P}_{\text{te}}(y = L \mid x) \cdot \mathbb{P}_{\text{te}}(y \neq L) \\
&\iff \frac{c_{\text{in}} \cdot \mathbb{P}_{\text{te}}(y = L)}{c_{\text{out}} \cdot \mathbb{P}_{\text{te}}(y \neq L)} < \frac{\mathbb{P}_{\text{te}}(y = L \mid x)}{\mathbb{P}_{\text{te}}(y \neq L \mid x)} \\
&\iff \mathbb{P}_{\text{te}}(y = L \mid x) > F\left( \frac{c_{\text{in}} \cdot \mathbb{P}_{\text{te}}(y = L)}{c_{\text{out}} \cdot \mathbb{P}_{\text{te}}(y \neq L)} \right).
\end{aligned}
$$

We further have for threshold $t^*_{\text{osc}} \doteq F\left( \frac{c_{\text{in}} \cdot \mathbb{P}_{\text{te}}(y=L)}{c_{\text{out}} \cdot \mathbb{P}_{\text{te}}(y \neq L)} \right)$,

$$
\begin{aligned}
\mathbb{P}_{\text{te}}(y = L \mid x) \geq t^*_{\text{osc}} &\iff \mathbb{P}_{\text{te}}(y \neq L \mid x) \leq 1 - t^*_{\text{osc}} \\
&\iff \frac{1}{\mathbb{P}_{\text{te}}(y \neq L \mid x)} \geq \frac{1}{1 - t^*_{\text{osc}}} \\
&\iff \frac{\max_{y' \neq L} \mathbb{P}_{\text{te}}(y' \mid x)}{\mathbb{P}_{\text{te}}(y \neq L \mid x)} \geq \frac{\max_{y' \neq L} \mathbb{P}_{\text{te}}(y' \mid x)}{1 - t^*_{\text{osc}}} \\
&\iff \max_{y' \neq L} \mathbb{P}_{\text{in}}(y' \mid x) \geq \frac{\max_{y' \neq L} \mathbb{P}_{\text{te}}(y' \mid x)}{1 - t^*_{\text{osc}}}.
\end{aligned}
$$

That is, we want to reject when the maximum softmax probability is *higher* than some (sample-dependent) threshold. □

*Proof of Lemma I.3.* Fix $\epsilon \in (0, 1)$. We consider two cases for threshold $t_{\text{msp}}$:

Case (i): $t_{\text{msp}} \leq \frac{1}{L-1}$. Consider a distribution where for all instances $x$, $\mathbb{P}_{\text{te}}(y = L \mid x) = 1 - \epsilon$ and $\mathbb{P}_{\text{te}}(y' \mid x) = \frac{\epsilon}{L-1}, \forall y' \neq L$. Then the Bayes-optimal classifier accepts any instance $x$ for all thresholds $t \in (0, 1 - \epsilon)$. In contrast, Chow's rule would compute $\max_{y \neq L} \mathbb{P}_{\text{in}}(y \mid x) = \frac{1}{L-1}$, and thus reject all instances $x$.

Case (ii): $t_{\text{msp}} > \frac{1}{L-1}$. Consider a distribution where for all instances $x$, $\mathbb{P}_{\text{te}}(y = L \mid x) = \epsilon$ and $\mathbb{P}_{\text{te}}(y' \mid x) = \frac{1-\epsilon}{L-1}, \forall y' \neq L$. Then the Bayes-optimal classifier would reject any instance $x$ for thresholds $t \in (\epsilon, 1)$, whereas Chow's rule would accept all instances.

Taking $\epsilon \to 0$ completes the proof. □

## I.2  ILLUSTRATION OF MSP FAILURE FOR OPEN-SET CLASSIFICATION

Figure 1 shows a graphical illustration of the example discussed in Example I.2, wherein the MSP baseline can fail for open-set classification. Figure 2 has another example setting.

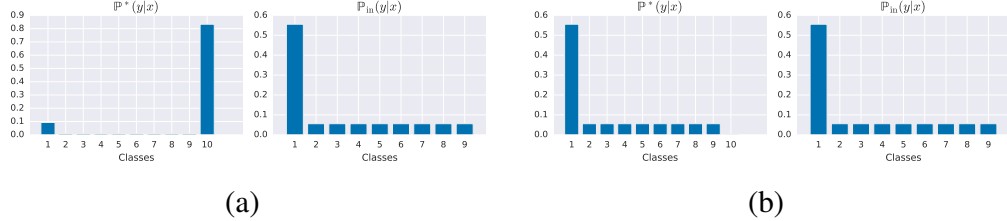

Figure 1: Examples of two open-set classification settings (a) and (b) with $L = 10$ classes, where the inlier class distributions $\mathbb{P}_{\text{in}}(y \mid x) = \frac{\mathbb{P}_{\text{te}}(y|x)}{\mathbb{P}_{\text{te}}(y \neq 10|x)}$ over the first 9 classes are identical, but the unknown class density $\mathbb{P}^*(10|x)$ is significantly different. Consequently, the MSP baseline, which relies only on the inlier class probabilities, will output the same rejection decision for both settings, whereas the Bayes-optimal classifier, which rejects by thresholding $\mathbb{P}^*(10|x)$, may output different decisions for the two settings.

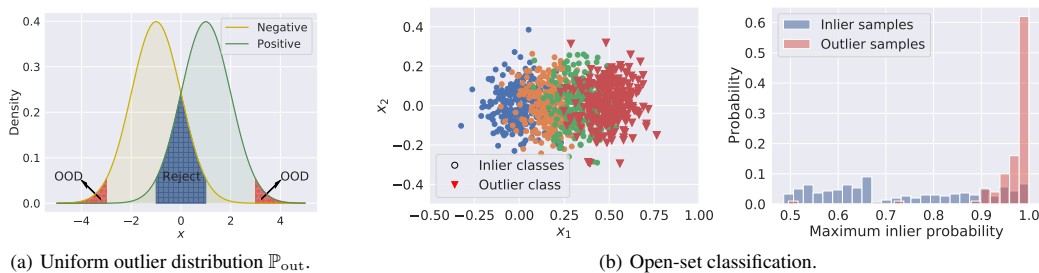

Figure 2: Example of two settings where the maximum softmax probability (MSP) baseline fails for OOD detection. Setting **(a)** considers *low-density OOD detection*, where positive and negative samples drawn from a one-dimensional Gaussian distribution. Samples *away* from the origin will have $\mathbb{P}(x) \sim 0$, and are thus outliers under the Bayes-optimal OOD detector. However, the MSP baseline will deem samples *near* the origin to be outliers, as these have maximal $\max_y \mathbb{P}(y \mid x)$. This illustrates the distinction between abstentions favoured by L2R (low label certainty) and OOD detection (low density). Setting **(b)** considers *open-set classification* where there are $L = 4$ total classes, with the fourth class (denoted by ▼) assumed to comprise outliers not seen during training. Each class-conditional is an isotropic Gaussian (left). Note that the maximum *inlier* class-probability $\mathbb{P}_{\text{in}}(y \mid x)$ scores OOD samples significantly *higher* than ID samples (right). Thus, the MSP baseline, which declares samples with low $\max_y \mathbb{P}_{\text{in}}(y \mid x)$ as outliers, will perform poorly.

### I.3 ILLUSTRATION OF MAXIMUM LOGIT FAILURE FOR OPEN-SET CLASSIFICATION

we show in Figure 3 the maximum logit computed over the inlier distribution. As with the maximum probability, the outlier samples tend to get a higher score than the inlier samples.

For the same reason, rejectors that threshold the margin between the highest and the second-highest probabilities, instead of the maximum class probability, can also fail. The use of other SC methods such as the cost-sensitive softmax cross-entropy (Mozannar and Sontag, 2020) may not be successful either, because the optimal solutions for these methods have the same form as MSP.

## J ADDITIONAL EXPERIMENTS

We provide details about the hyper-parameters and dataset splits used in the experiments, as well as, additional experimental results and plots that were not included in the main text. The in-training experimental results are **averaged over 5 random trials**.

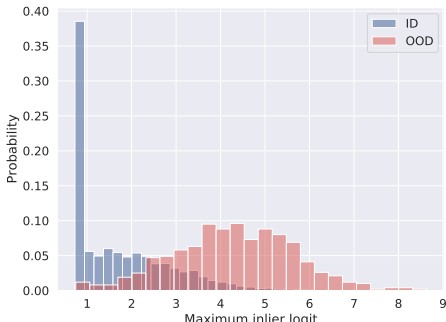

Figure 3: For the same setting as Figure 2, we show the maximum logit computed over the inlier distribution. As with the maximum probability, the outlier samples tend to get a higher score than the inlier samples.

| Rejector | Tunable Parameter | Estimated Parameter | Training Samples | Validation/Test Samples |
|---|---|---|---|---|
| Black-box (BB) (Tab. 2–4, 6–12) | $\lambda$ | - | ID samples | ID + OOD (te) samples |
| Loss-based (LB) (Tab. 2–3, 6–10) | $\lambda$ | $\pi_{\mathrm{mix}}$ | ID, Unlabeled mix of ID + OOD (tr), Strictly ID | ID + OOD (te) samples |

Table 5: Summary of hyper-parameters and dataset splits for different settings. We assume the practitioner specifies $c_{\mathrm{fn}}$ and $b_{\mathrm{rej}}$, and that $\pi_{\mathrm{in}}^*$ is known.

## J.1 HYPER-PARAMETER CHOICES

We provide details of the learning rate (LR) schedule and other hyper-parameters used in our experiments.

| Dataset | Model | LR | Schedule | Epochs | Batch size |
|---|---|---|---|---|---|
| CIFAR-40/100 | CIFAR ResNet 56 | 1.0 | anneal | 256 | 1024 |

We use SGD with momentum as the optimization algorithm for all models. For annealing schedule, the specified learning rate (LR) is the initial rate, which is then decayed by a factor of ten after each epoch in a specified list. For CIFAR, these epochs are 15, 96, 192 and 224.

Furthermore, as noted in §4.3, the proposed plug-in estimators requires specification of $c_{\mathrm{in}}$ and $c_{\mathrm{out}}$, which we are given by $c_{\mathrm{in}} = c_{\mathrm{fn}} - \lambda \cdot (1 - \pi_{\mathrm{in}}^*)$ and $c_{\mathrm{out}} = \lambda \cdot \pi_{\mathrm{in}}^*$, where $\lambda$ is a *tunable* parameter, and $\pi_{\mathrm{in}}^*$ is the proportion of ID samples in the test population, which we assume to be known. Interestingly, we find our plug-in estimators to be robust to the specification of this parameter. Table 5 summarizes the details for both the black-box (§4.1) and loss-based (§4.2) settings.

## J.2 BASELINE DETAILS

We provide further details about the baselines we compare with. The following baselines are trained on only the inlier data.

- *MSP or Chow's rule*: Train a scorer $f : \mathcal{X} \to \mathbb{R}^L$ using CE loss, and threshold the MSP to decide to abstain (Chow, 1970; Hendrycks and Gimpel, 2017).
- *MaxLogit*: Same as above, but instead threshold the maximum logit $\max_{y \in [L]} f_y(x)$ (Hendrickx et al., 2021).
- *Energy score*: Same as above, but threshold the energy function $-\log \sum_y \exp(f_y(x))$ (Liu et al., 2020a).
- *DOCTOR*: Same as above, but threshold the scorer $1 - \sum_y \mathrm{softmax}_y(f(x))^2$ (Granese et al., 2021).

Table 6: AUC-RC ($\downarrow$) for CIFAR-100 as ID, and a "wild" comprising of 90% ID and *only* 10% OOD. The OOD part of the wild set is drawn from the *same* OOD dataset from which the test set is drawn. We compare the proposed methods with the cost-sensitive softmax (CSS) learning-to-reject loss of Mozannar and Sontag (2020) and the ODIN method of Hendrickx et al. (2021). The test set contains 50% ID and 50% OOD samples. We set $c_{\text{fn}} = 0.75$.

| Method / $\mathbb{P}_{\text{out}}^{\text{te}}$ | ID + OOD training with $\mathbb{P}_{\text{out}}^{\text{tr}} = \mathbb{P}_{\text{out}}^{\text{te}}$ | | |
| --- | --- | --- | --- |
| | SVHN | Places | OpenImages |
| CSS | 0.286 | 0.263 | 0.254 |
| ODIN | 0.218 | 0.217 | **0.217** |
| Plug-in BB [$L_1$] | **0.196** | 0.210 | 0.222 |
| Plug-in BB [Res] | 0.198 | 0.236 | 0.251 |
| Plug-in LB* | 0.221 | **0.199** | 0.225 |

- *ODIN*: Train a scorer $f : \mathcal{X} \to \mathbb{R}^L$ using CE loss, and uses a combination of input noise and temperature-scaled MSP to decide when to abstain Hendrickx et al. (2021).
- *k-NN*: Train a scorer $f : \mathcal{X} \to \mathbb{R}^L$ using CE loss, compute embeddings from the embedding layer of the scorer, and threshold the (negative) 2-norm distance to the $k$-th nearest training sample in the embedding space (Sun et al., 2022).
- *SIRC*: Train a scorer $f : \mathcal{X} \to \mathbb{R}^L$ using CE loss, and compute a post-hoc deferral rule that combines the MSP score with either the $L_1$-norm or the residual score of the embedding layer from the scorer $f$ (Xia and Bouganis, 2022).
- *CSS*: Minimize the cost-sensitive softmax L2R loss of Mozannar and Sontag (2020) using only the inlier dataset to learn a scorer $f : \mathcal{X} \to \mathbb{R}^{L+1}$, augmented with a rejection score $f_\perp(x)$, and abstain iff $f_\perp(x) > \max_{y' \in [L]} f_{y'}(x) + t$, for threshold $t$.

The following baselines additional use the unlabeled data containing a mix of inlier and OOD samples.

- *Coupled CE (CCE)*: Train a scorer $f : \mathcal{X} \to \mathbb{R}^{L+1}$, augmented with a rejection score $f_\perp(x)$ by optimizing the CCE loss of Thulasidasan et al. (2021), and abstain iff $f_\perp(x) > \max_{y' \in [L]} f_{y'}(x) + t$, for threshold $t$.
- *De-coupled CE (DCE)*: Same as above but uses the DCE loss of Bitterwolf et al. (2022) for training.
- *Outlier Exposure (OE):* Train a scorer using the OE loss of Hendrycks et al. (2019) and threshold the MSP.

### J.3    DATA SPLIT DETAILS

For the CIFAR-100 experiments where we use a wild sample containing a mix of ID and OOD examples, we split the original CIFAR-100 training set into two halves, use one half as the inlier sample and the other half to construct the wild sample. For evaluation, we combine the orignal CIFAR-100 test set with the respective OOD test set. In each case, the larger of the ID and OOD dataset is down-sampled to match the desired ID-OOD ratio. The experimental results are **averaged over 5 random trials**.

For the pre-trained ImageNet experiments, we sample equal number of examples from the ImageNet validation sample and the OOD dataset, and annotate them with the pre-trained model. The number of samples is set to the smaller of the size of the OOD dataset or 5000.

### J.4    COMPARISON TO CSS AND ODIN BASELINES

We present some representative results in Table 6 comparing our proposed methods against the cost-sensitive softmax (CSS) of Mozannar and Sontag (2020), a representative learning-to-reject baseline, and the ODIN method of Hendrickx et al. (2021), an OOD detection baseline. As expected, the CSS baseline, which does not have OOD detection capabilities is seen to under-perform. The ODIN, baseline, on the other hand, is occasionally seen to be competitive.

Table 7: Area Under the Risk-Coverage Curve (AUC-RC) for methods trained with CIFAR-100 as the ID sample and a mix of CIFAR-100 and 300K Random Images as the wild sample, and with the proportion of OOD samples in test set varied. The wild set contains 10% ID and 90% OOD. The test sets contain 50% ID and 50% OOD samples. Base model is ResNet-56. $c_{\mathrm{fn}} = 0.75$. A * against a method indicates that it uses both ID and OOD samples for training. *Lower* values are *better*.

| Method / $\mathbb{P}_{\mathrm{out}}^{\mathrm{te}}$ | Test OOD proportion = 0.25 | | | | | Test OOD proportion = 0.75 | | | | |
|---|---|---|---|---|---|---|---|---|---|---|
| | SVHN | Places | LSUN | LSUN-R | Texture | SVHN | Places | LSUN | LSUN-R | Texture |
| MSP | 0.166 | 0.185 | 0.178 | 0.221 | 0.188 | 0.488 | 0.519 | 0.507 | 0.559 | 0.520 |
| MaxLogit | 0.154 | 0.183 | 0.166 | 0.211 | 0.181 | 0.461 | 0.507 | 0.488 | 0.544 | 0.509 |
| Energy | 0.156 | 0.183 | 0.169 | 0.211 | 0.185 | 0.462 | 0.508 | 0.489 | 0.542 | 0.511 |
| DOCTOR | 0.166 | 0.184 | 0.176 | 0.220 | 0.189 | 0.488 | 0.519 | 0.505 | 0.559 | 0.522 |
| SIRC [$L_1$] | 0.147 | 0.184 | 0.161 | 0.219 | 0.172 | 0.464 | 0.515 | 0.486 | 0.557 | 0.507 |
| SIRC [Res] | 0.133 | 0.183 | 0.155 | 0.219 | 0.166 | 0.442 | 0.516 | 0.477 | 0.555 | 0.494 |
| CCE* | 0.175 | 0.191 | 0.153 | 0.131 | 0.154 | 0.460 | 0.487 | 0.425 | 0.374 | 0.429 |
| DCE* | 0.182 | 0.200 | 0.155 | 0.136 | 0.162 | 0.467 | 0.498 | 0.414 | 0.372 | 0.428 |
| OE* | 0.179 | 0.174 | 0.147 | 0.117 | 0.148 | 0.492 | 0.487 | 0.440 | 0.371 | 0.440 |
| Plug-in BB [$L_1$] | 0.124 | 0.180 | 0.135 | 0.207 | 0.139 | 0.395 | 0.490 | **0.412** | 0.508 | **0.422** |
| Plug-in BB [Res] | **0.110** | 0.180 | 0.134 | 0.194 | 0.146 | **0.378** | 0.503 | 0.416 | 0.476 | 0.451 |
| Plug-in LB* | 0.160 | **0.169** | **0.133** | **0.099** | **0.132** | 0.468 | **0.489** | 0.418 | **0.351** | 0.430 |

| Method / $\mathbb{P}_{\mathrm{out}}^{\mathrm{te}}$ | Test OOD proportion = 0.01 | | | | | Test OOD proportion = 0.99 | | | | |
|---|---|---|---|---|---|---|---|---|---|---|
| | SVHN | Places | LSUN | LSUN-R | Texture | SVHN | Places | LSUN | LSUN-R | Texture |
| MSP | 0.063 | 0.064 | 0.063 | 0.065 | 0.063 | 0.731 | 0.732 | 0.731 | 0.734 | 0.733 |
| MaxLogit | 0.069 | 0.070 | 0.070 | 0.071 | 0.068 | 0.727 | 0.736 | 0.734 | 0.734 | 0.739 |
| Energy | 0.071 | 0.072 | 0.071 | 0.072 | 0.071 | 0.727 | 0.734 | 0.734 | 0.736 | 0.735 |
| DOCTOR | **0.062** | 0.064 | 0.063 | 0.065 | 0.063 | 0.730 | 0.731 | 0.731 | 0.733 | 0.733 |
| SIRC [$L_1$] | **0.062** | 0.063 | **0.062** | 0.064 | **0.062** | 0.728 | 0.731 | 0.731 | 0.735 | 0.730 |
| SIRC [Res] | **0.062** | 0.063 | **0.062** | 0.065 | **0.062** | 0.726 | 0.731 | 0.730 | 0.734 | 0.731 |
| CCE* | 0.105 | 0.106 | 0.104 | 0.103 | 0.105 | 0.727 | 0.735 | **0.724** | 0.715 | 0.727 |
| DCE* | 0.115 | 0.115 | 0.113 | 0.113 | 0.113 | 0.732 | 0.735 | **0.724** | 0.714 | 0.729 |
| OE* | 0.084 | 0.085 | 0.084 | 0.082 | 0.083 | 0.730 | **0.729** | 0.726 | 0.715 | **0.725** |
| Plug-in BB [$L_1$] | **0.062** | **0.062** | **0.062** | 0.065 | 0.063 | 0.722 | 0.733 | 0.725 | 0.731 | 0.728 |
| Plug-in BB [Res] | **0.062** | 0.064 | **0.062** | 0.065 | **0.062** | **0.719** | 0.735 | 0.727 | 0.728 | 0.731 |
| Plug-in LB* | 0.065 | 0.065 | 0.064 | **0.062** | **0.062** | 0.727 | **0.729** | **0.724** | **0.709** | **0.725** |

## J.5 VARYING OOD MIXING PROPORTION IN TEST SET

We repeat the experiments in Table 2 on CIFAR-100 and 100K Random Images with varying proportions of OOD samples in the test set, and present the results in Table 7. In each case, we assume that the proportion of OOD samples in the test set is known when computing $c_{\mathrm{in}}$ and $c_{\mathrm{out}}$ (§4.3), although we find our plug-in estimators to be robust to this parameter. We find one among the proposed plug-in methods continues to perform the best.

## J.6 VARYING OOD COST PARAMETER

We repeat the experiments in Table 2 on CIFAR-100 and 100K Random Images with varying values of cost parameter $c_{\mathrm{fn}}$, and present the results in Table 8. One among the proposed plug-in methods continues to perform the best. The lower the value of $c_{\mathrm{fn}}$, the closer the SCOD problem in (3) is to classical OOD detection (i.e., lower is the importance given to classification accuracy on inlier samples). When $c_{\mathrm{fn}} = 1$, the AUC-RC metric in (11) solely evaluates the quality of OOD detection (ignoring inlier classification performance).

Table 8: Area Under the Risk-Coverage Curve (AUC-RC) for methods trained with CIFAR-100 as the ID sample and a mix of CIFAR-100 and 300K Random Images as the wild sample, and for different values of cost parameter $c_{\text{fn}}$. The wild set contains 10% ID and 90% OOD. The test sets contain 50% ID and 50% OOD samples. Base model is ResNet-56.

| Method / $\mathbb{P}_{\text{out}}^{\text{te}}$ | $c_{\text{fn}} = 0.5$ | | | | | $c_{\text{fn}} = 0.9$ | | | | |
|---|---|---|---|---|---|---|---|---|---|---|
| | SVHN | Places | LSUN | LSUN-R | Texture | SVHN | Places | LSUN | LSUN-R | Texture |
| MSP | 0.256 | 0.271 | 0.265 | 0.297 | 0.275 | 0.336 | 0.376 | 0.358 | 0.442 | 0.381 |
| MaxLogit | 0.253 | 0.275 | 0.263 | 0.294 | 0.277 | 0.301 | 0.359 | 0.325 | 0.414 | 0.359 |
| Energy | 0.254 | 0.276 | 0.263 | 0.295 | 0.279 | 0.301 | 0.359 | 0.325 | 0.414 | 0.363 |
| DOCTOR | 0.255 | 0.271 | 0.263 | 0.296 | 0.273 | 0.335 | 0.376 | 0.357 | 0.440 | 0.381 |
| SIRC [$L_1$] | 0.248 | 0.271 | 0.259 | 0.296 | 0.267 | 0.300 | 0.372 | 0.324 | 0.438 | 0.346 |
| SIRC [Res] | 0.240 | 0.272 | 0.254 | 0.295 | 0.263 | 0.269 | 0.372 | 0.313 | 0.435 | 0.333 |
| CCE* | 0.296 | 0.307 | 0.283 | 0.269 | 0.286 | 0.282 | 0.318 | 0.233 | 0.179 | 0.240 |
| DCE* | 0.303 | 0.317 | 0.285 | 0.270 | 0.292 | 0.289 | 0.331 | 0.225 | 0.177 | 0.238 |
| OE* | 0.287 | 0.283 | 0.270 | 0.255 | 0.272 | 0.327 | **0.315** | 0.252 | 0.173 | 0.251 |
| Plug-in BB [$L_1$] | 0.237 | 0.270 | 0.244 | 0.289 | 0.248 | 0.208 | 0.333 | **0.223** | 0.358 | **0.237** |
| Plug-in BB [Res] | **0.232** | 0.271 | 0.244 | 0.279 | 0.255 | **0.187** | 0.347 | 0.225 | 0.305 | 0.270 |
| Plug-in LB* | 0.256 | **0.265** | **0.243** | **0.222** | **0.245** | 0.299 | 0.326 | 0.234 | **0.165** | 0.246 |

Table 9: Area Under the Risk-Coverage Curve (AUC-RC) for methods trained with CIFAR-100 as the ID sample and a mix of CIFAR-100 and 300K Random Images as the wild sample, with 95% **confidence intervals** included. The wild set contains 10% ID and 90% OOD. The test sets contain 50% ID and 50% OOD samples. Base model is ResNet-56. We set $c_{\text{fn}} = 0.75$.

| Method / $\mathbb{P}_{\text{out}}^{\text{te}}$ | SVHN | Places | LSUN | LSUN-R | Texture |
|---|---|---|---|---|---|
| MSP | $0.307 \pm 0.015$ | $0.335 \pm 0.017$ | $0.322 \pm 0.009$ | $0.387 \pm 0.027$ | $0.340 \pm 0.004$ |
| MaxLogit | $0.282 \pm 0.014$ | $0.327 \pm 0.015$ | $0.302 \pm 0.009$ | $0.368 \pm 0.030$ | $0.332 \pm 0.007$ |
| Energy | $0.282 \pm 0.013$ | $0.327 \pm 0.015$ | $0.300 \pm 0.010$ | $0.369 \pm 0.031$ | $0.329 \pm 0.007$ |
| DOCTOR | $0.305 \pm 0.014$ | $0.337 \pm 0.016$ | $0.324 \pm 0.008$ | $0.385 \pm 0.028$ | $0.341 \pm 0.004$ |
| SIRC [$L_1$] | $0.281 \pm 0.012$ | $0.334 \pm 0.018$ | $0.300 \pm 0.009$ | $0.385 \pm 0.028$ | $0.318 \pm 0.005$ |
| SIRC [Res] | $0.256 \pm 0.011$ | $0.336 \pm 0.018$ | $0.290 \pm 0.007$ | $0.382 \pm 0.028$ | $0.309 \pm 0.005$ |
| CCE* | $0.288 \pm 0.017$ | $0.315 \pm 0.018$ | $0.252 \pm 0.004$ | $0.213 \pm 0.001$ | $0.255 \pm 0.004$ |
| DCE* | $0.295 \pm 0.015$ | $0.326 \pm 0.028$ | $0.246 \pm 0.004$ | $0.212 \pm 0.001$ | $0.260 \pm 0.005$ |
| OE* | $0.313 \pm 0.015$ | **$0.304 \pm 0.006$** | $0.261 \pm 0.001$ | $0.204 \pm 0.002$ | $0.260 \pm 0.002$ |
| Plug-in BB [$L_1$] | $0.223 \pm 0.006$ | $0.318 \pm 0.025$ | $0.237 \pm 0.008$ | $0.351 \pm 0.040$ | **$0.244 \pm 0.004$** |
| Plug-in BB [Res] | **$0.205 \pm 0.002$** | $0.324 \pm 0.020$ | **$0.240 \pm 0.005$** | $0.319 \pm 0.026$ | $0.265 \pm 0.004$ |
| Plug-in LB* | $0.290 \pm 0.017$ | **$0.306 \pm 0.016$** | $0.243 \pm 0.003$ | **$0.186 \pm 0.001$** | $0.248 \pm 0.006$ |

## J.7 CONFIDENCE INTERVALS

In Table 9, we report 95% confidence intervals for the experiments on CIFAR-100 and 100K Random Images from Table 2. In each case, the differences between the best performing plug-in method and the baselines are *statistically significant*.

## J.8 COVARIATE-SHIFTED OOD SETTING

In Table 10, we present experimental results on a covariate-shifted OOD setting, where the ID dataset is CIFAR-100, and the OOD dataset we evaluate on during test time is a noise corrupted version of CIFAR-100 (Hendrycks and Dietterich, 2019; Tian et al., 2022), which we refer to as CIFAR-100-C. Since both ID and OOD samples being variants of the same dataset. this task is more challenging than the previous ones. We evaluate both methods that use only ID samples, and methods that are additionally provided images from Random300K, as a part of a "wild" dataset. The latter are seen to fare better, with our proposed loss-based plug-in method (Plug-in LB) performing the best.

Table 10: Area Under the Risk-Coverage Curve (AUC-RC) for methods trained with CIFAR-100 as the ID sample and a mix of CIFAR-100 and either 300K Random Images as the wild sample ($c_{\text{fn}} = 0.75$). The OOD dataset we evaluate on during test time is a version of CIFAR-100 corrupted by 15 types of noises (Hendrycks and Dietterich, 2019; Tian et al., 2022), referred to as CIFAR-100-C. The wild set contains 10% ID and 90% OOD. The test sets contain 50% ID and 50% OOD samples. Base model is ResNet-56. A * against a method indicates that it uses both ID and OOD samples for training. *Lower* values are *better*.

| Method | CIFAR-100-C |
|---|---|
| MSP | 0.359 |
| MaxLogit | 0.356 |
| Energy | 0.355 |
| DOCTOR | 0.361 |
| SIRC [$L_1$] | 0.357 |
| SIRC [Res] | 0.355 |
| CCE* | 0.212 |
| DCE* | 0.214 |
| OE* | 0.204 |
| Plug-in BB [$L_1$] | 0.357 |
| Plug-in BB [Res] | 0.343 |
| Plug-in LB* | **0.185** |

## J.9 ADDITIONAL RESULTS ON PRE-TRAINED IMAGENET MODELS

Following Xia and Bouganis (2022), we present additional results with pre-trained models with ImageNet-200 (a subset of ImageNet with 200 classes) as the inlier dataset in Table 11. The base model is a ResNet-50. In Table 12, we once again present our experiments on ImageNet with the BiT ResNet-101 base model, with additional comparisons with the nearest-neighbor scorers of Sun et al. (2022).

## K LIMITATIONS AND BROADER IMPACT

Recall that our proposed plug-in rejectors seek to optimize for overall classification and OOD detection accuracy while keeping the total fraction of abstentions within a limit. However, the improved overall accuracy may come at the cost of poorer performance on smaller sub-groups. For example, Jones et al. (2021) show that Chow's rule or the MSP scorer "can magnify existing accuracy disparities between various groups within a population, especially in the presence of spurious correlations". It would be of interest to carry out a similar study with the two plug-in based rejectors proposed in this paper, and to understand how both their inlier classification accuracy and their OOD detection performance varies across sub-groups. It would also be of interest to explore variants of our proposed rejectors that mitigate such disparities among sub-groups.

Another limitation of our proposed plug-in rejectors is that they are only as good as the estimators we use for the density ratio $\frac{\mathbb{P}_{\text{in}}(x)}{\mathbb{P}_{\text{out}}(x)}$. When our estimates of the density ratio are not accurate, the plug-in rejectors are seen to often perform worse than the SIRC baseline that use the same estimates. Exploring better ways for estimating the density ratio is an important direction for future work.

Beyond SCOD, the proposed rejection strategies are also applicable to the growing literature on adaptive inference Liu et al. (2020a). With the wide adoption of large-scale machine learning models with billions of parameters, it is becoming increasingly important that we are able to perform speed up the inference time for these models. To this end, adaptive inference strategies have gained popularity, wherein one varies the amount of compute the model spends on an example, by for example, exiting early on "easy" examples. The proposed approaches for SCOD may be adapted to equip early-exit models to not only exit early on high-confidence "easy" samples, but also exit early on samples that are deemed to be outliers. In the future, it would be interesting to explore the design of such early-exit models that are equipped with an OOD detector to aid in their routing decisions.

Table 11: AUC-RC (↓) for methods trained with ImageNet-200 as the inlier dataset and *without* OOD samples. The base model is a pre-trained ResNet-50 model. *Lower* values are *better*.

| Method / $\mathbb{P}_{out}^{te}$ | Places | LSUN | CelebA | Colorectal | iNaturalist-O | Texture | ImageNet-O | Food32 |
|---|---|---|---|---|---|---|---|---|
| | | | | | ID-only training | | | |
| MSP | 0.183 | 0.186 | 0.156 | 0.163 | 0.161 | 0.172 | 0.217 | 0.181 |
| MaxLogit | **0.173** | 0.184 | 0.146 | 0.149 | 0.166 | 0.162 | 0.209 | 0.218 |
| Energy | 0.176 | 0.185 | 0.145 | 0.146 | 0.172 | 0.166 | 0.211 | 0.225 |
| DOCTOR | 0.179 | 0.185 | 0.152 | 0.155 | 0.159 | 0.170 | 0.226 | 0.175 |
| NN ($k = 1$) | 0.234 | 0.234 | 0.186 | **0.136** | 0.253 | 0.154 | 0.199 | 0.263 |
| NN ($k = 5$) | 0.239 | 0.252 | 0.171 | 0.139 | 0.222 | **0.143** | 0.204 | 0.285 |
| NN ($k = 10$) | 0.252 | 0.284 | 0.177 | 0.140 | 0.230 | 0.148 | **0.184** | 0.323 |
| SIRC [$L_1$] | 0.185 | 0.195 | 0.155 | 0.165 | 0.166 | 0.172 | 0.214 | 0.184 |
| SIRC [Res] | 0.180 | 0.179 | 0.137 | 0.140 | 0.151 | 0.167 | 0.219 | 0.174 |
| Plug-in BB [$L_1$] | 0.262 | 0.261 | 0.199 | 0.225 | 0.228 | 0.270 | 0.298 | 0.240 |
| Plug-in BB [Res] | 0.184 | **0.172** | **0.135** | **0.138** | **0.145** | 0.194 | 0.285 | **0.164** |

| Method / $\mathbb{P}_{out}^{te}$ | Near-ImageNet-200 | Caltech65 | Places32 | Noise |
|---|---|---|---|---|
| | ID-only training | | | |
| MSP | 0.209 | 0.184 | 0.176 | 0.188 |
| MaxLogit | 0.220 | **0.171** | **0.170** | 0.192 |
| Energy | 0.217 | 0.175 | **0.169** | 0.190 |
| DOCTOR | **0.198** | **0.170** | 0.171 | 0.187 |
| NN ($k = 1$) | 0.252 | 0.182 | 0.232 | 0.139 |
| NN ($k = 5$) | 0.280 | 0.182 | 0.227 | 0.141 |
| NN ($k = 10$) | 0.295 | 0.190 | 0.249 | 0.140 |
| SIRC [$L_1$] | 0.205 | 0.182 | 0.174 | 0.191 |
| SIRC [Res] | 0.204 | 0.177 | 0.173 | **0.136** |
| Plug-in BB [$L_1$] | 0.264 | 0.242 | 0.256 | 0.344 |
| Plug-in BB [Res] | 0.247 | 0.202 | **0.171** | **0.136** |

Table 12: AUC-RC (↓) for methods trained with ImageNet-200 as the inlier dataset and *without* OOD samples. The base model is a pre-trained ResNet-50 model. *Lower* values are *better*.

| Method / $\mathbb{P}_{out}^{te}$ | ID-only training | | | | | | | |
| | Places | LSUN | CelebA | Colorectal | iNaturalist-O | Texture | ImageNet-O | Food32 |
|---|---|---|---|---|---|---|---|---|
| MSP | 0.183 | 0.186 | 0.156 | 0.163 | 0.161 | 0.172 | 0.217 | 0.181 |
| MaxLogit | **0.173** | 0.184 | 0.146 | 0.149 | 0.166 | 0.162 | 0.209 | 0.218 |
| Energy | 0.176 | 0.185 | 0.145 | 0.146 | 0.172 | 0.166 | 0.211 | 0.225 |
| DOCTOR | 0.179 | 0.185 | 0.152 | 0.155 | 0.159 | 0.170 | 0.226 | 0.175 |
| NN ($k = 1$) | 0.234 | 0.234 | 0.186 | **0.136** | 0.253 | 0.154 | 0.199 | 0.263 |
| NN ($k = 5$) | 0.239 | 0.252 | 0.171 | 0.139 | 0.222 | **0.143** | 0.204 | 0.285 |
| NN ($k = 10$) | 0.252 | 0.284 | 0.177 | 0.140 | 0.230 | 0.148 | **0.184** | 0.323 |
| SIRC [$L_1$] | 0.185 | 0.195 | 0.155 | 0.165 | 0.166 | 0.172 | 0.214 | 0.184 |
| SIRC [Res] | 0.180 | 0.179 | 0.137 | 0.140 | 0.151 | 0.167 | 0.219 | 0.174 |
| Plug-in BB [$L_1$] | 0.262 | 0.261 | 0.199 | 0.225 | 0.228 | 0.270 | 0.298 | 0.240 |
| Plug-in BB [Res] | 0.184 | **0.172** | **0.135** | **0.138** | **0.145** | 0.194 | 0.285 | **0.164** |

| Method / $\mathbb{P}_{out}^{te}$ | ID-only training | | | |
| | Near-ImageNet-200 | Caltech65 | Places32 | Noise |
|---|---|---|---|---|
| MSP | 0.209 | 0.184 | 0.176 | 0.188 |
| MaxLogit | 0.220 | **0.171** | **0.170** | 0.192 |
| Energy | 0.217 | 0.175 | **0.169** | 0.190 |
| DOCTOR | **0.198** | **0.170** | 0.171 | 0.187 |
| NN ($k = 1$) | 0.252 | 0.182 | 0.232 | 0.139 |
| NN ($k = 5$) | 0.280 | 0.182 | 0.227 | 0.141 |
| NN ($k = 10$) | 0.295 | 0.190 | 0.249 | 0.140 |
| SIRC [$L_1$] | 0.205 | 0.182 | 0.174 | 0.191 |
| SIRC [Res] | 0.204 | 0.177 | 0.173 | **0.136** |
| Plug-in BB [$L_1$] | 0.264 | 0.242 | 0.256 | 0.344 |
| Plug-in BB [Res] | 0.247 | 0.202 | **0.171** | **0.136** |

