# OpenReview forum: "Plugin estimators for selective classification with out-of-distribution detection"
_ICLR.cc/2024/Conference — ICLR 2024 poster_

### Official Review · Reviewer_WNPt · 2023-10-27

**Soundness:** 2 fair
**Presentation:** 3 good
**Contribution:** 3 good
**Rating:** 6
**Confidence:** 4

**Summary:**

In this paper, the authors explore a novel setting, selective classification with OOD detection (SCOD), which combines two common settings in machine learning, selective classification (SC) and out-of-distribution (OOD) detection. This investigation hold promise, as recent studies have highlighted the propensity for challenging in-distribution (ID) samples to be mistakenly classified as OOD. The authors provide a theoretical foundation for understanding SCOD, and they introduce a method that adeptly integrates existing SC and OOD detection techniques. The experimental results affirm that the proposed method performs admirably in SCOD settings.

**Strengths:**

1. The integration of SC and OOD in the setting is intriguing, as it more closely resembles real-world scenarios.
2. This paper is well-presented, providing a detailed certification process.
3. The proposed method demonstrates superior results in both settings.

**Weaknesses:**

1. How to choose hyperparameters is crucial for this combined method, and it would be beneficial if the authors could provide more detail.
2. Evaluating performance on datasets that are exclusively OOD or SC is meaningful, as the proportion of each may vary across different scenarios.
3. The performance in the absence of OOD samples is not outstanding, with some results even falling below those of the baselines.

**Questions:**

1. How to determine the PI_in* through inspection of logged data?

---

> ### Author Response · Authors · 2023-11-21
> **Response to Reviewer WNPt**
>
> We thank the reviewer for the encouraging comments and questions.
>
> > How to choose hyperparameters is crucial for this combined method, and it would be beneficial if the authors could provide more detail.
>
> Our method requires the specification of two parameters $c_{\rm in}$ and $c_{\rm out}$. As noted in Section 4.3, these are in turn chosen based on the abstention budget $b_{\rm rej}$ and the cost of non-rejection $c_{\rm fn}$ provided by the practitioner.
>
> Specifically, for Lagrange multiplier $\lambda$, we may expand $c_{\rm out} = c_{\rm fn} - \lambda \cdot (1 - \pi^*_{\rm in})$ and $c_{\rm in} = \lambda \cdot \pi^*_{\rm in}$, where $\pi^*_{\rm in}$ is the proportion of ID samples we expect in the test population.
>
> The Lagrange multiplier $\lambda$ is the only hyper-parameter that needs to be tuned. We tune  $\lambda$ so that the resulting rejector has a rejection rate of $b_{\rm rej}$ on a held-out set. Table 5 in the appendix summarizes the tuned and derived parameters.
>
> > How to determine the $\pi^*_{\rm in}$ through inspection of logged data?
>
> Sorry for the confusion. We simply meant that if one has a sample of historical data, it could be possible to perform manual labelling on a subset to identify the proportion of OOD samples, or equally, $1 - \pi^{*}_{\rm in}$.
>
> > Evaluating performance on datasets that are exclusively OOD or SC is meaningful, as the proportion of each may vary across different scenarios.
>
> **In Table 7, we include results for two extreme settings**, where the test set has 1% OOD samples and 99% OOD samples (to avoid numerical overflow issues with the metrics, we retain a small fraction of ID/OOD samples in each case).
>
> When all (or most) of the samples are ID, we find that the MSP method becomes very competitive, as this is known to be a strong baseline for selective classification tasks.
>
> When all (or most) of the samples are OOD, all the methods yield similar performance. This is because the optimal rejection strategy in this case is to declare almost all samples as OOD, and rejecting fewer samples is bound to incur a high cost. So irrespective of the method used, the AUC-RC metric, which averages the risk over different rejection rates, yields the same value.

---

### Official Review · Reviewer_ZjGh · 2023-10-27

**Soundness:** 3 good
**Presentation:** 3 good
**Contribution:** 3 good
**Rating:** 8
**Confidence:** 3

**Summary:**

This work proposes a new view of the framework for selective classification and out-of-distribution (SCOD) detection into a unified classifier, rejector tuple problem. Their framework allows for plugging in any existing OOD similarity score to perform rejection under different scenarios of data access, e.g., only in-distribution or a mixture of unlabeled IND and OOD data. They also showcase a statistical formulation for the problem and derive the Bayes optimal solution from it. They provide an empirical evaluation of the popular image classification benchmarks.

**Strengths:**

An extensive theoretical formulation of the SCOD problem, with derivations for the optimal classifier rejector pair and an alternative surrogate loss.

The experimental suite is large, with experiments on both CIFAR and ImageNet datasets, comparing to previous SCOD methods (SIRC) and traditional OOD detection metrics, but not selective classification methods.

**Weaknesses:**

* No new insights are given on better estimating the prediction confidence or the probability ratio between in and out distributions.

* Since no guarantees can be drawn for either $s_{ood}$ or $s_{sc}$, (8) or lemma 4.1 does not inherit any further guarantees.

* Loss-based results are only shown for the CIFAR benchmark, not ImageNet.

* The AUC-RC is compared against OOD methods but not against selective classification/misclassification detection methods such as [1], [2], [3], etc. Adding them to the benchmark would further strengthen the experiments.

**Notation**: sometimes the notation is a little bit confusing. $L$ is the number of classes, the loss function, and the Lagrangian. Also, $[\cdot]$ is a set, but {${\cdot}$} is also a set.

**Typo**: the second term of the surrogate loss in Algorithm 1 differs from the one in (10) (sampling over (x,y) instead of x). Notation might become heavy, but maybe introducing the marginal could be helpful. In algorithm 1, line 5 is not a probability if $\hat{s}$ maps into $\mathbb{R}$.

References:

[1] SelectiveNet. Geifman, Y. "SelectiveNet: A Deep Neural Network with an Integrated Reject Option." ICML 2019. /abs/1901.09192.\
[2] ConfidNet. Corbière et al. "Addressing Failure Prediction by Learning Model Confidence." NeurIPS 2019. /abs/1910.04851.\
[3] Doctor. Granes et. al. "DOCTOR: A Simple Method for Detecting Misclassification Errors." NeurIPS 2021. /abs/2106.02395.

**Questions:**

1. Could the authors show the steps to go from (3) to (4) to make $c_{in}$ and $c_{out}$ appear as in 4.3?
2. Could the excessive loss bound in Lemma 4.1 be rewritten by considering the estimation error of $P_{in}(y|x)$, $\pi^*_{in}$, and $\hat{s}_{ood}$?
3. The constraint considered in (3) takes into account the rejection rate on the test distribution. Why not consider the rejection rate only on the in-distribution like (1) and (2)?
4. Assumption (A2) in page 6 states that $P_{out}(x) = 0$ for $x$ in $S_{in}^{\*}$. How to guarantee this in practice? I.e., how to build $S_{in}^{\*}$? The proposed strategy in the footnote considers $(x,y)$ and not simply $x$. The way I see it is that it is impossible to obtain a strict $S^*_{in}$ without full knowledge on $P_{out}$.
5.  For CIFAR, the proposed SCOD learning in algorithm 1 does not seem to yield better results than training only on the CE loss and using heuristic scores to perform SCOD. Could the authors elaborate on potential limitations on why this is the case?
6. How does the inlier rejection option perform on this task compared to the proposed method and existing OOD detectors? Please check the references cited in the Weakness section.

---

> ### Author Response · Authors · 2023-11-21
> **Response to Reviewer ZjGh [Part 1]**
>
> We thank the reviewer for the detailed comments and questions.
>
> > Could the authors show the steps to go from (3) to (4)
>
> We have updated Appendix G with a more detailed derivation going from Equation 3 to Equation 4. See also Appendix B, which provides an argument for the appropriateness of solving the Lagrangian under mild distributional assumptions.
>
> > Could the excessive loss bound in Lemma 4.1 be rewritten by considering the estimation error?
>
> Thanks for the suggestion. In this setting, we assume access to existing SC and OOD detection scorers, and we bound the SCOD error in terms of how well these scores do on their respective tasks. Importantly, we don’t make specific assumptions about how these scores are obtained.
>
> If however we made more assumptions (e.g., we assume that they are the result of performing ERM with a certain surrogate loss), one could have an analogue to the generalisation analysis in Appendix C, which further bounds the estimation error of the individual scores. We will add a comment on this.
>
> > The constraint considered in (3) takes into account the rejection rate on the test distribution. Why not consider the rejection rate only on the in-distribution like (1) and (2)?
>
> This is a reasonable question. Appendix F discusses a potential alternative formulation for SCOD which indeed considers a constraint on the in-distribution. Note that this does not change the general form of the Bayes-optimal scorer, and thus does not change the general plug-in SCOD strategy. Indeed, this is equivalent to (3) with a particular (distribution-dependent) choice of constraint $b_{\rm rej}$.
>
> Regarding why constraining the rejection rate on the test distribution can make more sense: we are motivated by scenarios where the practitioner allocates a budget of the total fraction of abstentions they are comfortable making on test data. These abstentions could _either_ be due to samples being close to the decision boundary, _or_ due to samples being OOD.
>
> This formulation is natural in applications where an abstention  results in the sample being deferred to a larger model or a human expert at an additional monetary cost. In this case, the *abstention typically incurs the same cost whether it was an ID or OOD sample that was rejected*, and it is critical that the total proportion of abstentions be within a fixed budget
>
> > Ensuring Assumption (A2) in page 6 is satisfied
>
> In the case of image classification with a fixed set of classes (as considered in Footnote 1), the proposal is to collect “canonical” samples for each class (e.g., unambiguous cat images, unambiguous dog images). Since out-of-distribution samples are, by definition, those drawn from a wholly distinct distribution than the training sample, it should be the case that $\mathbb{P}_{\rm out}(x) = 0$ for such “canonical” samples.
>
> > For CIFAR, the proposed SCOD learning in algorithm 1 does not seem to yield better results than training only on the CE loss and using heuristic scores to perform SCOD. Could the authors elaborate on potential limitations on why this is the case?
>
> We believe the reviewer is referring to Table 2, where the **OOD samples seen during training are different from those used during test time**. In this case, our loss-based estimators (Algorithm 1) may sometimes perform worse than our black-box estimators, despite the latter being trained only on ID samples. This is primarily due to the mismatch between the OOD datasets used during training and testing.
>
> The reviewer’s observation does not however hold with the CIFAR100 experiments in **Table 3**, where we use samples from the *same* OOD dataset during both training and testing (albeit a part of a noisy “wild” dataset). Here, Algorithm 1 (last row) is seen to provide substantial improvements in 4 out 6 cases.
>
> ### Suggestions
>
> > How does the inlier rejection option perform on this task compared to the proposed method and existing OOD detectors?... The AUC-RC is compared against OOD methods but not against selective classification methods
>
> Please note that in Appendix I.4, we provided a comparison against a representative baseline from the SC literature, namely, the cost-sensitive softmax cross-entropy loss of (Mozannar and Sontag, 2020). We find similar conclusions to the results reported in the body. Additionally, the MSP method we compare against in all tables is the same as Chow’s rule in the SC literature (Chow, 1970).
>
> In the interest of further expanding the comparison against SC methods, **we have added results for the selective classification method of DOCTOR** method of Granese et al. (2021) [3] in **Tables 2, 3 and 13**.  This method yields similar performance as the MSP, MaxLogit and Energy scorers that use a base model trained with ID samples alone.

---

> > ### Author Response · Authors · 2023-11-21
> > **Response to Reviewer ZjGh [Part 2]**
> >
> > > Typo: the second term of the surrogate loss in Algorithm 1 differs from the one in (10) (sampling over (x,y) instead of x).
> >
> > Sorry for the confusion. We believe this equation is correct as stated: one loops over the samples in $S_{\rm in}$ – each of which is a pair $(x, y)$ – and then computes the loss based on the first element in the pair, i.e., $x$.
> >
> > We understand the reviewer’s suggestion to be defining a new set, say $X_{\rm in} = \\{ x \colon (x, y) \in S_{\rm in} \\}$. This would be equally precise, but we felt the notation might get too heavy. We are happy to reconsider if the reviewer believes the provided notation is confusing.
> >
> > > In algorithm 1, line 5 is not a probability if $\hat{s}$ maps to $\mathbb{R}$
> >
> > The reviewer is correct that $\hat{s}$ maps to $\mathbb{R}$ and not a probability estimate. Since $\hat{s}$ is trained using a binary cross-entropy loss $\ell_{\rm bc}(z, s(x)) = \log(1 + e^{-z \cdot s(x)})$, we may apply a **sigmoid** transformation to $\hat{s}(x)$ to obtain an estimate for $\frac{\\mathbb{P}\_{\rm in}( x )}{\\mathbb{P}\_{\rm in}( x ) + \\mathbb{P}\_{\rm mix}( x )}$, or exponentiate $-s(x)$ to get an estimate for $\frac{ \\mathbb{P}\_{\rm mix}( x ) }{ \\mathbb{P}\_{\rm in}( x ) }$. We provide details in Appendix E.

---

> > > ### Comment · Reviewer_ZjGh · 2023-11-22
> > >
> > > I thank the author's for addressing all my questions and acknowledge their effort put into the rebuttal. All my concerns were satisfactorily answered and I raise my score accordingly.

---

> > > > ### Author Response · Authors · 2023-11-22
> > > > **Thank you!**
> > > >
> > > > We are happy to know your concerns were satisfactorily addressed! Appreciate you raising your score. Thanks for the encouraging and helpful feedback.

---

### Official Review · Reviewer_nGxu · 2023-11-07

**Soundness:** 4 excellent
**Presentation:** 3 good
**Contribution:** 4 excellent
**Rating:** 8
**Confidence:** 4

**Summary:**

The paper addresses the problem of selective classification with OOD detection (SCOD), where the goal is to learn a classifier, rejector pair that can abstain on “hard” in-distribution samples as well as OOD samples. The unification of the selective classification (SC) and OOD detection areas has recently been explored, but its formal underpinnings have not been fully developed. The paper presents a formal statistical analysis of SCOD and derives the Bayes-optimal solution for the classifier, rejector pair. This solution generalizes the Bayes-optimal solutions for the SC and OOD detection problems considered separately. Based on this solution, they propose plug-in estimators for optimally combining confidence scores for SC and density-ratio scores for OOD detection. This provides a principled way of combining existing scoring methods from the SC and OOD detection areas to address the SCOD problem. They address two settings, the first one being a black-box setting with only in-distribution training data, and the second one being a loss-based setting where one additionally has access to an unlabeled mixture of in-distribution and OOD data.

**Strengths:**

- Strong theoretical work that unifies the seemingly disparate literatures of selective classification and OOD detection. The proposed statistical formulation and Bayes optimal solution for the classifier and rejector is a general result that can guide the design of selective classifiers that can reject on both uncertain in-distribution (mis-classified) inputs and OOD inputs. Although the areas of selective classification and OOD detection have been independently studied well, a combined analysis of the problems in a principled setting has been lacking and somewhat heuristic. This paper addresses the gap.

- The proposed plug-in estimators specify how to optimally combine existing confidence scores from the selective classification literature and density-ratio scores from the OOD detection literature. Therefore, it allows researchers in these areas to leverage, in a principled way, existing scoring methods for selective classification and OOD detection. The novelty of the paper does not lie in new approaches to estimate the confidence scores or the density-ratio scores, but rather in how to optimally combine them for the SCOD problem.

- They also propose a loss-based approach for learning the classifier and rejector by leveraging an unlabeled mixture of in-distribution and OOD data “in the wild” (similar to Katz-Samuels et al., 2022).

- Overall, the paper was interesting and insightful to read. There is a lot of discussion and results in the appendices which could be useful to researchers in this area.

**Weaknesses:**

1. The experiments mainly focus on semantic OOD (or far OOD) inputs, where there is no intersection in the label space of the in-distribution and OOD. It is also important to consider covariate-shifted OOD, e.g. which are caused due to common corruptions, noise weather changes etc. Some results on the covariate-shifted OOD data would strengthen the paper.

2. The method requires a few constants or hyper-parameters to be set. For instance, the cost of false negatives $c_{fn}$, the maximum rejection rate $b_{rej}$, the proportion of inlier and OOD data in the unlabeled set $S_{mix}$, the choice of training OOD data $P^{tr}_{out}$. The results in the main paper are for specific choices of these parameters (understably due to the page limit), but there is not much discussion or takeaways on how these parameters affect the performance. Some discussion on this would be useful.

3. Minor: there is lack of clarity in some parts of the paper, which could be improved. Please see the Questions section.

4. The code has not been made available and some implementation details are missing.

**Questions:**

### 1. Conditional probability?
In the formulation of selective classification, would it not be better to use the conditional probability of misclassification given the input is accepted, i.e. $P_{in}(y \neq h(x) \~|\~ r(x) = 0)$? Also, under the subsection `Evaluation metrics` on page 8, the joint risk is divided by the total number of accepted inputs, which seems to be consistent with the conditional probability.

### 2. On the Lagrangian
It is not clear to me how to arrive at the Lagrangian in Eqn (4) from the objective (3). Based on my simplification of the Lagrangian from the SCOD objective (3), I am getting a slightly different form than that in Eqn (4). Specifically, I get $c\_{in} = \lambda \pi^{\star}\_{in}$ and $c\_{out} = c\_{fn} - \lambda (1 - \pi^{\star}\_{in})$. Referring to Section 4.3, it seems like $c_{in}$ and $c_{out}$ specified here may have been swapped?

Also, it seems to me the multiplier of the first term in Eqn (4) is $(1 \~-\~ c\_{out} \~-\~ ((1 - \pi^{\star}\_{in}) / \pi^{\star}\_{in}) \\, c\_{in})$, rather than $1 - c\_{in} - c\_{out}$.

It is certainly possible I missed/messed something, but would appreciate some clarification.

### 3. Need for strictly-inlier dataset
It seems to me that the strictly-inlier dataset $S^{\star}\_{in}$ is only needed for estimating the mixture proportion $\hat{\pi}\_{mix}$. It is not clear why the labeled inlier dataset $S_{in}$ (with the labels discarded) cannot be used in place of $S^{\star}\_{in}$ for estimating $\hat{\pi}_{mix}$. Any theoretical reason for this?

### 4. Estimation of $\pi^{\star}\_{in}$
Please clarify if $\pi^{\star}\_{in}$ used in the formulation for SCOD Eqn (3) can be estimated using $\hat{\pi}_{mix}$? Does it have to be estimated at test time as mentioned in Footnote 3 on page 7? From my understanding, the mixture dataset $S\_{mix}$ is collected “in the wild” during deployment, which should give a good idea of $\pi^{\star}\_{in}$ as well.

### Suggestions for the Theorems/Lemmas
- Would help to restate the Lemma/Theorem statements in the appendix.
- Some comments/takeaways on Lemma 4.1 and Lemma 4.2 would be useful.
- In Lemma 3.1, I believe it should be: Let $(h^\star, r^\star)$ denote any minimiser of (3) (not (2)).
- In the proof of Lemma 3.1 (Appendix A), it seems like the reject class $\perp$ is allowed to be part of the classifier $h(x)$ output. However, the classifier is originally defined as $h : \mathcal{X} \mapsto [L]$. Please clarify this point.
- Some pointers could be added to the proofs. For example, Eqn (13) on page 18 follows from Pinsker’s inequality, which is worth mentioning.
- Lemma 4.2: it should be $s^\star$ not $r^\star$.
- Lemma 4.2: it would be useful to clarify that $p_{\perp}(x)$ is an approximation for $P^\star(z = 1 \~|\~ x)$. The $\perp$ symbol is commonly used for rejection, whereas here it corresponds to the probability of accepting.
- Lemma 4.2: for introducing coupling implicitly it should be $s(x) = u^T \Phi(x)$ (no dependency on $y’$ here).
- Typo in the first line of the Proof of Lemma 4.2 on page 17: it should be $s^\star(x) = \log(P^\star(z=1 | x) / P^\star(z=-1 | x))$.

### Rejector definition
Might be better to define the rejectors $r^\star(x)$ and $r_{BB}(x)$ in Eqn (5) and Eqn (8) directly using the indicator function $\mathbb{1}[\cdot]$.

### On the Algorithm
- Line 3 of Algorithm 1: should it be $\hat{f} : \mathcal{X} \mapsto \mathrm{R}^L$? That is, $\hat{f}$ predicts a logit for each of the $L$ classes.
- Line 7 of Algorithm 1: Can directly specify the final classifier $\hat{h}(x) = \arg\max_{y} \hat{f}_y(x)$ since it does not depend on Eqn (8). Can also specify that $s\_{sc}(x) = \max\_{y} \hat{f}_y(x)$ in Eqn (8).

### Other points
1. Under `Baselines` on page 8: reference for energy-based scorer should be (Liu et al., 2020b) and (Hendrycks & Gimpel, 2017) should also be cited for MSP.

2. In Table 2, why do methods such as MSP, MaxLogit, and Energy (which do not use OOD training data) have different performance under the two settings: $P^{tr}\_{out}=$ Random300K and $P^{tr}\_{out}=$ OpenImages?

3. From Table 4, it seems that the performance of `Plug-in BB [L_1]` is consistently worse than `SIRC [L_1]`, despite the former (proposed) method being more principled. Please explain this discrepancy.

4. I think it would be useful to provide results of this method using the Deep NN method (Sun et al., 2022), especially on the ImageNet dataset, since it seems like the grad-norm scorers are not providing good estimates of the density ratio.

5. It might be worth citing the following paper which characterizes the Bayes-optimal detector for mis-classification detection.
Doctor: A simple method for detecting misclassification errors, https://proceedings.neurips.cc/paper_files/paper/2021/hash/2cb6b10338a7fc4117a80da24b582060-Abstract.html

---

> ### Author Response · Authors · 2023-11-21
> **Response to Reviewer nGxu [Part 1]**
>
> We warmly thank the reviewer for their encouraging comments and detailed feedback.
>
>
> > In the formulation of selective classification, would it not be better to use the conditional probability of misclassification given the input is accepted
>
> Indeed, there are two closely related formulations for classification problems with an abstention option: one uses the conditional error $\mathbb{P}(y \neq h(x) \mid r(x) = 0)$ (as exemplified in, e.g., SelectiveNet), while the other uses the joint error $\mathbb{P}(y \neq h(x), r(x) = 0)$ (as exemplified in, e.g., Ramaswamy et al. 2018).
>
> There is a one-to-correspondence between the two formulations: owing to the constraint on $\mathbb{P}(r(x) = 1)$ in Equation 1, it is not hard to see that
> $$min_{h, r}~ \mathbb{P}(y \neq h(x) \mid r(x) = 0) \colon \~ \mathbb{P}(r(x) = 1) \leq b$$
> $$min_{h, r}~ \frac{\mathbb{P}(y \neq h(x), r(x) = 0)}{P(r(x) = 0)} \colon \~ \mathbb{P}(r(x) = 1) \leq b$$
> $$min_{h, r, a}~ \frac{\mathbb{P}(y \neq h(x), r(x) = 0)}{a} \colon \~ \mathbb{P}(r(x) = 1) \leq b,\~ \mathbb{P}(r(x) = 0) \geq a$$
> $$min_{h, r, a}~ \frac{\mathbb{P}(y \neq h(x), r(x) = 0)}{a} \colon \~ \mathbb{P}(r(x) = 1) \leq \min(b, 1 - a)$$
>
> Thus, for a fixed a, the problem is equivalent to Equation 1, with a modified choice of the constraint on $\mathbb{P}(r(x) = 1)$. The Bayes-optimal classifier is thus unaffected.
>
> We agree that discussion of this point is worthwhile, and have added this to Appendix F. We note also that Appendix F had a discussion of a slightly different SCOD formulation.
>
> > It is not clear to me how to arrive at the Lagrangian in Eqn (4) from the objective (3)... it seems like $c_{\rm in}$ and $c_{\rm out}$ specified here may have been swapped?
>
> Thanks for the catch! The reviewer is correct: there was a typo with the constants $c_{\rm in}, c_{\rm out}$ being swapped. This has been fixed. We have also updated Appendix G with a more detailed derivation of going from Equation 3 to Equation 4, and updated the discussion in Section 4.3.
>
> > Need for strictly-inlier dataset
>
> The reviewer is correct that the strictly-inlier dataset $S^*_{\rm in}$ is only needed to estimate the mixing weight $\pi_{\rm mix}$.
>
> Per Lemma E.1, we use the assumption that $\\mathbb{P}\_{\rm out}(x) = 0$ for $x \in S^*_{\rm in}$ to estimate $\pi_{\rm mix}$. Note that if we just used $x \in S_{\rm in}$, we may not necessarily have $\\mathbb{P}\_{\rm out}(x) = 0$: the latter would only hold under the assumption that the support of $\\mathbb{P}\_{\rm in}$ and $\\mathbb{P}\_{\rm out}$ are disjoint. We allow for the two distributions to have overlapping support in general (e.g., outliers might correspond to draws from a low (but non-zero) density region of $\\mathbb{P}\_{\rm in}$).
>
> > Please clarify if $\pi^*_{\rm in}$ used in the formulation for SCOD Eqn (3) can be estimated using $\hat{\pi}_{mix}$?
>
> $\pi^*_{\rm in}$ is the proportion of inliers in the test set, while $\pi_{\rm mix}$ is the proportion in the wild set. As the reviewer points out, indeed $\pi^*_{\rm in}$ may be equal to $\pi_{\rm mix}$ in practice, as the wild set is drawn from a production system.
>
> In our experiments, we additionally consider cases where $\pi^*_{\rm in}$ can be different from $\pi_{\rm mix}$; in such cases, we will have to obtain estimates $\pi^*_{\rm in}$ through other means, such as through the practitioner or through manual inspection of historical data.
>
> > In the proof of Lemma 3.1 (Appendix A), it seems like the reject class  is allowed to be part of the classifier  output. Please clarify this point.
>
> Sorry for the confusion. One can either work with a classifier-rejector pair $(h, r)$, or an augmented classifier $\bar{h} \colon \mathcal{X} \to [ L ] \cup \{ \perp \}$. We have updated the proof of Lemma 3.1 to rewrite statements involving $h(x) = \perp$ to $r(x) = 1$.
>
> > In Table 2, why do methods such as MSP, MaxLogit, and Energy (which do not use OOD training data) have different performance under the two settings: Random300K and OpenImages?
>
> We thank the reviewer for spotting this discrepancy between the two sets of results. This difference is due to a subtle implementation detail that we had overlooked.
>
> All our models are trained on the same dataset, with each batch comprising both ID and OOD samples. When training the base model for MSP, MaxLogit, and Energy, we compute the loss only on the ID samples from a batch, and ignore the OOD samples. However, because the base ResNet model may internally perform other operations such as batch normalization that, for example, compute averages across the entire batch of ID + OOD samples, we find that the trained models still have a mild dependence on the OOD samples in the training data.
>
> We have now re-run the MSP, MaxLogit, and Energy results in Tables 2 and 3 on a dataset consisting exclusively of only ID samples. **The conclusions remain the same**, with one or more of the proposed plug-in methods often performing better than these baselines.

---

> > ### Author Response · Authors · 2023-11-21
> > **Response to Reviewer nGxu [Part 2]**
> >
> > > From Table 4, it seems that the performance of Plug-in BB [L_1] is consistently worse than SIRC [L_1], despite the former (proposed) method being more principled. Please explain this discrepancy.
> >
> > The proposed Plug-in BB method relies on being able to obtain reasonable estimates for the density ratio $\frac{\\mathbb{P}\_{\rm in}( x )}{\\mathbb{P}\_{\rm out}( x )}$. We attribute the discrepancy in Table 4 to the $L_1$-norm scorer being a poor estimate of the density ratio.
> >
> > More specifically, as shown in Lemma 3.1, the proposed method computes a weighted combination of the maximum softmax probability (MSP) and the density ratio estimate. Based on the weights assigned (which in turn depend on the rejection budget and practitioner specified costs), the scorer could be susceptible to errors in the density ratio estimates.
> >
> > In contrast, the SIRC [$L_1$] heuristic in equation (9) has a multiplicative form, without explicit specification of costs. Note that while this makes this heuristic less susceptible to errors in the density ratio estimates, it often performs near-identical to the MSP scorer (compare rows 1 and 4 in Table 4).
> >
> > Furthermore, when we do have a good estimate of the density ratio, as with the Residual-norm estimator (last row of Table 4), the proposed approach is often able to take better advantage of the improved density estimates compared to the SIRC heuristic.
> >
> >
> > ### Suggestions
> >
> > > I think it would be useful to provide results of this method using the Deep NN method (Sun et al., 2022), especially on the ImageNet dataset, since it seems like the grad-norm scorers are not providing good estimates of the density ratio.
> >
> > We include **comparisons to the Deep NN** method of Sun et al. (2022) on the ImageNet tasks in **Table 13** (Appendix I.12). In three out of 12 cases, one the NN methods fares the best. In the remaining, Plug-in BB [Res] is often either competitive to or better than the other methods.
> >
> > Sun et al. do not explicitly provide a way of measuring density ratio estimates. As with the grad-norm scorers, we could apply a transform to the NN-distance scores to estimate density ratios. We will include an exploration of this in the next version of the paper.
> >
> > > It might be worth citing the following paper which characterizes the Bayes-optimal detector for mis-classification detection.
> >
> > Thanks for pointing out this relevant work! We have added a citation in Section 2, as well as the Conclusion.
> >
> > > Rejector definition using indicator function
> >
> > Thanks for the suggestion, we have used the indicator notation to define these quantities.
> >
> > > Line 3 of Algorithm 1: should it be $\hat{f} \colon \mathcal{X} \to \mathbb{R}^L$?
> >
> > Thanks for the catch: indeed this is a typo, which we have fixed.
> >
> > > Under Baselines on page 8: reference for energy-based scorer should be (Liu et al., 2020b) and (Hendrycks & Gimpel, 2017) should also be cited for MSP.
> >
> > Thanks for the catch, we have fixed this.
> >
> > > Suggestions for the Theorems/Lemmas
> >
> > We have incorporated the reviewer’s many helpful suggestions, which are greatly appreciated. We will be happy to add the main takeaways from Lemma 4.1 and 4.2 in the main text.
> >
> > ### Other comments
> >
> > >  It is also important to consider covariate-shifted OOD, e.g. which are caused due to common corruptions ..
> >
> > We have included an additional experiment in Appendix I.9 (**Table 10**) on a **covariate-shifted OOD setting**. Here the ID dataset is CIFAR-100, and the OOD dataset we evaluate on during test time is **CIFAR-100-C**, a corrupted version of CIFAR-100 [Hendrycks, & Dietterich, 2019].
> >
> > We evaluate both methods that train on ID samples, and methods that use both ID samples and samples from the 300K Random Images dataset. The latter methods are seen to perform substantially better, with our proposed Plug-in LB approach performing the best. This shows that in these covariate shift settings, exposing the methods to some OOD images, even if not part of the original CIFAR-100-C dataset, can substantially improve their performance.
> >
> > > The code has not been made available
> >
> > We will certainly endeavour to release the code in the future. In the meantime, if there are some implementation details that would be useful to add but are currently missing, do let us know.

---

> > > ### Comment · Reviewer_nGxu · 2023-11-22
> > >
> > > Thank you for the detailed responses and revision of the paper with additional results. My concerns have been addressed.

---

> ### Author Response · Authors · 2023-11-22
> **Thank you!**
>
> We are glad your concerns were satisfactorily addressed! Thanks for the positive and encouraging feedback.

---

### Author Response · Authors · 2023-11-21
**Common response to all reviewers**

We thank the reviewers for their detailed comments! We have provided responses to specific questions in the individual reviewer threads.

We have also uploaded a revised version of the paper, based on the reviewer comments. The changes (highlighted in blue) are:
- Updated Appendix G with a more detailed derivation of the Lagrangian
- Added comparisons to DOCTOR method [Granese et al., 2021] suggested by Reviewer ZjGh (Tables 2, 3, 13)
- Added comparisons to Deep NN method of Sun et al. (2022) suggested by Reviewer nGxu (Tables 13)
- Added experiments on a covariate-shifted OOD setting suggested by Reviewer nGxu (Tables 10)
- Added experiments with an exclusive-SC and an exclusive-OOD setting suggested by Reviewer WNPt (Table 7)
- Fixed typos pointed out by Reviewer nGxu and ZjGh

---

### Meta-Review · Area_Chair_4ysd · 2023-12-05

**Metareview:**

The paper addresses the problem of selective classification with OOD detection (SCOD), where the goal is to learn a classifier, rejector pair that can abstain on “hard” in-distribution samples as well as OOD samples. The unification of the selective classification (SC) and OOD detection areas has recently been explored, but its formal underpinnings have not been fully developed. The paper presents a formal statistical analysis of SCOD and derives the Bayes-optimal solution for the classifier, rejector pair. This solution generalizes the Bayes-optimal solutions for the SC and OOD detection problems considered separately. Based on this solution, they propose plug-in estimators for optimally combining confidence scores for SC and density-ratio scores for OOD detection. This provides a principled way of combining existing scoring methods from the SC and OOD detection areas to address the SCOD problem. They address two settings, the first one being a black-box setting with only in-distribution training data, and the second one being a loss-based setting where one additionally has access to an unlabeled mixture of in-distribution and OOD data.

The clarity and novelty are clearly above the bar of ICLR. While the reviewers had some concerns on the experiments, such as hyperparameters selection and performance evaluation, the authors did a particularly good job in their rebuttal. Thus, all of us have agreed to accept this paper for publication! There is one minor point to be noted. Please include sufficient discussion with highly related works [1,2], because they also belong to post-hoc methods in the area of OOD detection. Please include the additional discussion and experimental results in the next version.

Reference:
[1] X Jiang et al. Detecting Out-of-distribution Data through In-distribution Class Prior. In ICML, 2023.
[2] Q Wang et al. Watermarking for Out-of-distribution Detection. In NeurIPS, 2022.

**Justification For Why Not Higher Score:**

The clarity and novelty are clearly above the bar of ICLR. While the reviewers had some concerns on the experiments, such as hyperparameters selection and performance evaluation, the authors did a particularly good job in their rebuttal. Thus, all of us have agreed to accept this paper for publication! There is one minor point to be noted. Please include sufficient discussion with highly related works [1,2], because they also belong to post-hoc methods in the area of OOD detection. Please include the additional discussion and experimental results in the next version.

Reference:
[1] X Jiang et al. Detecting Out-of-distribution Data through In-distribution Class Prior. In ICML, 2023.
[2] Q Wang et al. Watermarking for Out-of-distribution Detection. In NeurIPS, 2022.

**Justification For Why Not Lower Score:**

The clarity and novelty are clearly above the bar of ICLR. While the reviewers had some concerns on the experiments, such as hyperparameters selection and performance evaluation, the authors did a particularly good job in their rebuttal. Thus, all of us have agreed to accept this paper for publication! There is one minor point to be noted. Please include sufficient discussion with highly related works [1,2], because they also belong to post-hoc methods in the area of OOD detection. Please include the additional discussion and experimental results in the next version.

Reference:
[1] X Jiang et al. Detecting Out-of-distribution Data through In-distribution Class Prior. In ICML, 2023.
[2] Q Wang et al. Watermarking for Out-of-distribution Detection. In NeurIPS, 2022.

---

### Decision · Program_Chairs · 2024-01-16

Accept (poster)